# SUMMARYMIXING: A LINEAR-COMPLEXITY ALTERNATIVE TO SELF-ATTENTION FOR SPEECH RECOGNITION AND UNDERSTANDING

## ABSTRACT

Modern speech processing systems rely on self-attention. Unfortunately, token mixing with self-attention takes quadratic time in the length of the speech utterance, slowing down inference as well as training and increasing memory consumption. Cheaper alternatives to self-attention for ASR have been developed, but they fail to consistently reach the same level of accuracy. This paper, therefore, proposes a novel linear-time alternative to self-attention. It summarises an utterance with the mean over vectors for all time steps. This single summary is then combined with time-specific information. We call this method "SummaryMixing". Introducing SummaryMixing in state-of-the-art ASR models makes it feasible to preserve or exceed previous speech recognition performance while lowering the training and inference times by up to 28% and reducing the memory budget by a factor of two. The benefits of SummaryMixing can also be generalized to other speech-processing tasks, such as speech understanding.

## 1 INTRODUCTION

Automatic speech recognition (ASR) has greatly benefited from deep learning, reaching unprecedented levels of accuracy and enabling a number of successful products to support real-life use cases (Nassif et al., 2019; Arasteh et al., 2016). ASR systems have followed the general trend in deep learning and increased steadily both in model size and complexity to push recognition accuracies. Modern industry-scale ASR models often contain hundreds of millions or even billions of neural parameters (Radford et al., 2022). Training these models, however, often requires high amount of GPU hours and results in large carbon footprint (Parcollet & Ravanelli, 2021). Resource-efficient deployment and fast inference are also important factors, especially for on-device scenarios (Arasteh et al., 2016). Thus, this paper focuses on improving the efficiency of speech processing models.

At the core of current state-of-the-art (SOTA) speech systems are multi-head self-attention (MHSA) cells (Vaswani et al., 2017b). MHSA learns interactions between pairs of frames originating from the speech signal, and the interaction is also referred to as token mixing. MHSA has helped to reach SOTA performance in most speech models (Gulati et al., 2020; Peng et al., 2022; Baevski et al., 2020). On the other hand, considering each pair of frames takes quadratic time in the input sequence length. MHSA is therefore costly, particularly for long sequences.

Recent works pointed out that under some conditions, expensive pair-wise self-attention operations may behave like linear operations for SOTA speech recognition models. For example, Zhang et al. (2021b) first showed that the upper encoder layers in trained Transformer-based ASR models behave like feed-forward layers, which is also verified by Shim et al. (2022) for Conformer models. Furthermore, Peng et al. (2022) demonstrated that the attention matrices of trained Branchformer models tend to have diagonality scores (Zhang et al., 2021b) near 0.5. The definition of diagonality scores by Zhang et al. (2021b) implies that a 0.5 diagonality score means that the attention weights are distributed uniformly, and therefore, that a simple average could achieve a similar token mixing.

Therefore, inspired by HyperMixer (Mai et al., 2023) this work introduces a linear-time alternative to self-attention not relying on pair-wise interactions. Instead, it summarises a whole utterance as a mean over separate contributions for all time steps. The obtained summary is then fed back to each individual time step. We call this method "SummaryMixing".

Our proposed SummaryMixing[1] achieves a training time reduction of up to 28% and exhibits a stable real-time factor compared to a quadratic increase in latency for MHSA with utterance length. SummaryMixing also halves the memory consumption at training and decoding times in comparison with MHSA, and reaches the performance of SOTA ASR systems on five datasets of different languages and acoustic conditions (Section 3). These findings are extended to other speech understanding tasks including spoken language understanding (SLU) and keyword spotting (KWS). To the best of our knowledge, it is the first time a linear-time method matches or surpasses the performance of MHSA for speech-processing tasks across various scenarios.

## 1.1 RELATED WORK

Numerous existing efficient attention mechanisms attempt to re-create the original behavior of self-attention but at a lower training cost. For instance, active research directions include low-rank approximation (Tay et al., 2022), linearization (Wang et al., 2020), or sparsification (Child et al., 2019) of self-attention. In the context of ASR, the squeezeformer (Kim et al., 2022), the efficient conformer (Burchi & Vielzeuf, 2021) and the emformer (Shi et al., 2021) obtain lower training times and memory consumption than models equipped with the standard self-attention by reducing the length of the sequence attended without decreasing the attention span to avoid ASR performance degradation. Such approaches do not overcome the quadratic time complexity of MHSA as they only change the input length to limit or control the growth in complexity. This issue is approached by locality-biased linear attention introduced by Sun et al. (2022) or linear nyström attention by Samarakoon & Leung (2022). In these methods, the time complexity of self-attention is reduced to linear for ASR. Unfortunately, none of these solutions is able to reach SOTA ASR performance. Fastformer (Wu et al., 2021) is a successful linear alternative to self-attention as it has demonstrated superior performance to MHSA and a wide range of linear alternatives including Longformer (Beltagy et al., 2020), BigBird (Zaheer et al., 2020), Linformer (Wang et al., 2020) and Poolingformer (Zhang et al., 2021a) on standard natural language processing tasks. Fastformer was also extended to speech recognition in the Branchformer architecture (Peng et al., 2022) and, despite achieving faster training times, achieved lower ASR performance than MHSA. However, Fastformer remains the best-performing linear alternative to self-attention whose implementation is also available to the community and will be used as a baseline. SummaryMixing hypothesizes that the pair-wise information modeling capabilities of self-attention can be simplified and captured in a more efficient way. ContextNet (Han et al., 2020), the best-performing entirely convolutional ASR system, follows the same assumption and does not rely on MHSA to reach true SOTA performance. Unfortunately, no existing open-source implementation, even from major toolkits, is able to reproduce the reported ASR results. Also, the training time is not significantly better than MHSA-equipped ASR models. Despite these issues, ContextNext remains the best available non-attentional end-to-end ASR model.

A recently proposed method called the HyperMixer (Mai et al., 2023) derives from the MLP Mixer (Tolstikhin et al., 2021). The MLP Mixer (Tolstikhin et al., 2021) was the first to show that token mixing can also be achieved outside the framework of self-attention. MLP Mixer learns a fixed-size MLP to perform token mixing throughout time and achieves competitive performance across a wide range of domains with linear time complexity (Choe et al., 2022). Unfortunately, speech tasks commonly have variable-length sequences and existing extensions of the MLP Mixer to this modality only propose to rely on chunked and fixed-length input vectors and achieve sub-optimal performance (Sakuma et al., 2021). HyperMixer (Mai et al., 2023), on the other hand, is an attempt to extend MLP Mixer to varying time dimensions in the input space for natural language processing. Though it is not clear from the original presentation, the HyperMixer has the same time complexity as SummaryMixing, and a similar general architecture. Appendix A.1 shows this in detail.

## 2 SUMMARYMIXING

Previous works (Zhang et al., 2021b; Peng et al., 2022) have shown that speech recognition does not necessarily require long-distance fine-grained modeling at the acoustic level to reach state-of-the-art performance. Hence, this section will first introduce SummaryMixing (section 2.1) and how it is integrated into the Branchformer and Conformer architectures (section 2.2).

---

[1]The source code is available at `anonymised`.

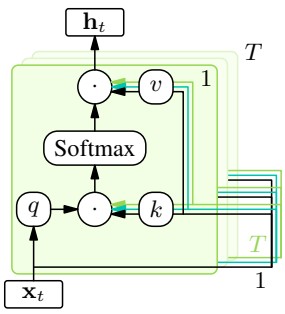 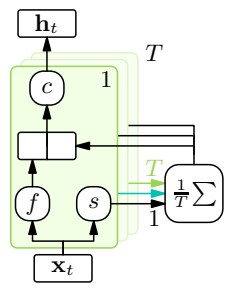 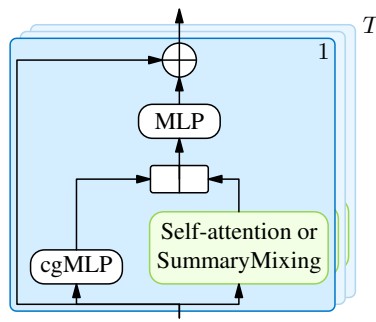

(a) The self-attention cell. Each pair of inputs is compared with each other input, requiring a quadratic number of operations.

(b) The newly proposed SummaryMixing cell. Information from all time steps is averaged, and this average is fed back to each time step $T$.

(c) The Branchformer. It has two branches: a convolutional branch for local information, and an attention branch for global information.

Figure 1: The main neural architectures in this work. The green block inside the Branchformer in Fig. 1c is implemented as self-attention (Fig. 1a) or SummaryMixing (Fig. 1b).

## 2.1 SUMMARYMIXING

Figure 1a shows a self-attention cell (Vaswani et al., 2017a). The plate and its content are replicated for each time step $t = 1 \ldots T$. The cell takes an input sequence $\mathbf{X} \in \mathbb{R}^{T \times D} = \{\mathbf{x}_0, \ldots, \mathbf{x}_T\}$ of $T$ feature vectors $\mathbf{x}_t$ of length $D$, and transforms them into hidden representations $\mathbf{H} \in \mathbb{R}^{T \times D'} = \{\mathbf{h}_0, \ldots, \mathbf{h}_T\}$, that can be inputs for the next layer. The self-attention cell computes a weighted average, where the weights are computed for each time step, of "values" (computed with $v$) for each time step. The multitude of connections across time steps in Figure 1a highlight the quadratic cost in the length of the input that self-attention takes.

To reduce the quadratic time complexity, we introduce SummaryMixing, which also transforms input vectors $\mathbf{x}_t$ into hidden representations $\mathbf{h}_t$. The key to inducing linear time complexity is to summarise the whole utterance in a single vector $\bar{\mathbf{s}}$. Figure 1b illustrates this. The input $\mathbf{x}_t$ is transformed by two functions. One is the local transformation function $f : \mathbb{R}^D \to \mathbb{R}^{D''}$. The other is summary function $s : \mathbb{R}^D \to \mathbb{R}^{D'''}$. The resulting vectors $s(\mathbf{x}_t)$ are averaged across all time steps ($\frac{1}{T}\sum$) to form the mean vector $\bar{\mathbf{s}}$. This single vector is passed back to each time step. The concatenation of it and the local information $f(\mathbf{x}_t)$ is then transformed by the combiner function $c : \mathbb{R}^{D''} + \mathbb{R}^{D'''} \to \mathbb{R}^{D'}$.

Mathematically, the SummaryMixing process can be described as:

$$\bar{\mathbf{s}} = \frac{1}{T} \sum_{t=1}^{T} s(\mathbf{x}_t); \qquad \mathbf{h}_t = c(f(\mathbf{x}_t), \bar{\mathbf{s}}). \qquad (1)$$

Each output vector is the function of one vector capturing the whole sequence and one capturing local information. Computing $\bar{\mathbf{s}}$ takes $\mathcal{O}(T)$ time, after which each $\mathbf{h}_t$ can be computed in constant time w.r.t. $T$. This compares to $\mathcal{O}(T^2)$ in standard self-attention.

**Relationship to the HyperMixer.** The HyperMixer was proposed by Mai et al. (2023) as a more efficient alternative for self-attention. Though the original description does not make it explicit, its time complexity w.r.t. the input length is linear, like SummaryMixing. Its overall layout is also similar, but it fixes the form of some functions. In contrast, in SummaryMixing, we implement $f$, $s$ and $c$ as general neural networks. Appendix A.1 gives more detail.

## 2.2 THE BRANCHFORMER AND CONFORMER WITH SUMMARYMIXING

The Branchformer (Peng et al., 2022) and Conformer (Gulati et al., 2020) reach state-of-the-art speech recognition and understanding accuracy. Figure 1c illustrates the structure of the Branchformer, with a place (in green) where normally self-attention goes. For our proposed method, self-attention is replaced with SummaryMixing. The same may be applied to the Conformer as depicted by the corresponding figure in the Appendix A.3.

In particular, the transformation ($f$), summary ($s$), and combiner ($c$) functions are all implemented as a dense linear layer followed by a GeLU activation function. The input of the combiner is a concatenation of $\bar{s}$ and $f(\mathbf{x}_t)$. The CNN branch of the Branchformer is an MLP with convolutional gating inspired by the cgMLP of (Sakuma et al., 2021). The convolutional gating helps in capturing strong local relations as the kernel size typically is limited to a small number of frames. The CNN modules of the Conformer play a similar role but are computed sequentially with the MHSA blocks instead of in parallel as in the Branchformer. In the case of the Branchformer, the outputs of both branches, CNN and SummaryMixing, are then concatenated and fed to a two-layered MLP followed by GeLU activations before feeding into the next block. For the Conformer, the output of the SummaryMixing block is simply fed to the next convolutional module. No layer normalization is applied within the SummaryMixing cell and a skip connection connects the input to the output after fusion from the last MLP.

**Branchformer with SummaryMixing-lite.** We propose one additional variation of the Branchformer combined with SummaryMixing. This variation is specific to this architecture and can not be extended to the Conformer. Instead of implementing the green block in Figure 1c as a SummaryMixing cell, SummaryMixing can be fully integrated. This uses the "cgMLP" block for the local transformation $f$, and the "MLP" block for the combiner function $c$. The summary function $s$ and the sum take the place of the green block. This is illustrated graphically in Appendix A.3. We hypothesize that SummaryMixing-lite speeds up training even further and reduces the memory footprint. However, a slight degradation in performance may be encountered as SummaryMixing-lite has lost its per-time-step non-linear transformation, while the cgMLP offers a contextualized representation filtering over multiple frames that may loose some local information.

## 3 EXPERIMENTS

SummaryMixing is evaluated in a two-step process based on different experimental setups (Section 3.1). First, empirical efficiency gains in terms of training speed, memory consumption as well as real-time decoding factor (RTF) are highlighted in controlled tasks (Section 3.2). Then, the compared models are scaled to standard automatic speech recognition (ASR) and spoken language understanding (SLU) evaluations with common datasets and architectures (Section 3.3).

### 3.1 EXPERIMENTAL PROTOCOL

Different hyperparameters, architectures, and datasets are described in the corresponding sections while the baselines and the framework are shared and described here.

**Baseline architectures.** All ASR and SLU models, except those stated otherwise, are based on the encoder-decoder architecture (Karita et al., 2019). For ASR, decoders are either following the joint CTC/Attention training scheme (Kim et al., 2017) or CTC only (Graves & Graves, 2012). The attentional decoder always is a standard Transformer whose parameters are detailed in the Appendix. The CTC decoder is a simple linear layer of the size of the output vocabulary which varies depending on the dataset and the task. For classification tasks, such as keyword spotting, no decoder is used. The output representations of the encoder are averaged over time and then classified by a dense neural network whose output dimension is equal to the number of classes. Multi-head self-attention is implemented in SpeechBrain and based on either relative positional encoding (Dai et al., 2019) with a custom PyTorch implementation or regular MHSA based on the official PyTorch implementation (i.e. CuDNN). The choice of the method is based on empirical results based on the task. Models are compared by varying the encoder architecture based on the literature.

More precisely, we first consider a standard Transformer architecture as it remains simple to comprehend but also achieves competitive performance (Karita et al., 2019). Then, we consider the Conformer from Gulati et al. (2020) as it currently is employed in most deployed ASR systems from real-world products (Guo et al., 2021). The Branchformer is included as our main baseline as it is an improvement over the Conformer (Peng et al., 2022). The E-Branchformer (Kim et al., 2023), however, is not added as it does not represent a single model, but instead, a class of different architectures that are extremely close performance and architecture-wise to the Branchformer. We then consider a Branchformer equipped with the FastFormer (Wu et al., 2021)

attention mechanism as a baseline for a SOTA linear complexity self-attention (Peng et al., 2022). Indeed, Fastformer has been shown to outperform the best linear alternatives to self-attention in various tasks (Wu et al., 2021). We also introduce a Branchformer equipped with HyperMixer attention (Mai et al., 2023) to highlight the performance improvements compared to this token mixing technique. We integrate two CNN-only models with ContextNet, first introduced by Han et al. (2020), being considered for the competitive baseline. ContextNet remains, to this day, the best-performing CNN-only architecture for ASR according to the original results. It is worth emphasizing that no open-source implementation of ContextNet is able to reproduce the originally reported results. Finally, and to serve as a low bar baseline, a single branch Branchformer is proposed by removing the MHSA part, hence leading to a CNN-only Branchformer. This will be particularly useful to quantify the impact of the global context obtained from MHSA or other alternatives.

**Implementation details.** ASR and SLU systems have been implemented within the widely used and open-source SpeechBrain toolkit (Ravanelli et al., 2021) version *0.5.14*. Experiments are conducted following officially available and open-source recipes with only the architecture of the models being changed. The latter allows for a fair comparison as the training procedure and environment are strictly equivalent, well adopted within the speech community, and therefore directly comparable and replicable. The extended list of hyperparameters for each experiment is available in the Appendix A.6 and in the code repository. Reported results are obtained from re-training and not from the literature.

## 3.2 EFFICIENCY AND REAL-TIME FACTOR ANALYSIS

These controlled experiments are expected to highlight the growth in training time, decoding speed, and VRAM requirements as a function of the input utterance length.

**Efficiency task details.** Models are comprised of an encoder, described thereafter for each candidate, and a simple linear layer for a decoder trained with CTC. No autoregressive decoder is used to avoid any side effects induced by the use of random input vectors and labels. Systems are benchmarked both in terms of measured training time, i.e., a combination of the time necessary to execute the forward and backward pass as well as the neural parameters update, and peak VRAM consumption across the entire task. In practice, five thousand sequences of random tensors corresponding to a signal of length $L$ with $1 \leq L \leq 100$ in seconds and sampled at 16 kHz are generated and used as inputs, while one hundred random tokens corresponding to indices in a vocabulary of size 1,000 are used as targets. The final training time is an average of the 5,000 sentences and is expressed in seconds. Measurements are extracted on an isolated compute node with four Tesla A100 80GB. Bfloat16 mixed precision (Kalamkar et al., 2019) is enabled to replicate the real training conditions.

**Real-time factor task details.** Real Time Factor (RTF) measurements are obtained by dividing the time taken to decode an utterance by its duration. Hence, all models pre-trained on Librispeech, whose WER results are reported in Table 1, are compared on the observed RTF by varying the length of the utterances to decode. In particular, we constructed six sets of 2,500 sentences of duration $10, 20, 30, 40, 50, 60$ seconds. All sentences in a set have the exact same duration from the *test-clean* of the Librispeech dataset. Hence, they are either cropped to the corresponding duration or concatenated to reach longer lengths (typically $30, 40, 60$ seconds). Models are compared on the exact same sets of sentences to avoid any variation. Measurements are obtained from a batched greedy CTC decoding. The batch size is set to 16 to avoid exceeding the available amount of VRAM for the Branchformer model. Measurements are from an isolated node with a Tesla A100 80GB.

**Model architectures.** The selected models for these experiments are the Branchformer equipped with MHSA, FastFormer linear attention, SummaryMixing and SummaryMixing-lite as well as ContextNet. For the efficiency analysis, the number of neural parameters is set to roughly 80M for the SummaryMixing Branchformer and vanilla Branchformer, and 65M for the SummaryMixing-lite Branchformer and ContextNet. This corresponds to architectures reaching state-of-the-art WER on Librispeech without the Transformer decoder. The SummaryMixing-lite Branchformer and ContextNet networks are slightly smaller since they replace attention with a simple linear projection. The Branchformers share the same internal dimension between blocks, 512, as well as the number of encoder blocks, 18. In practice, all parameters between both sets of models are identical, except

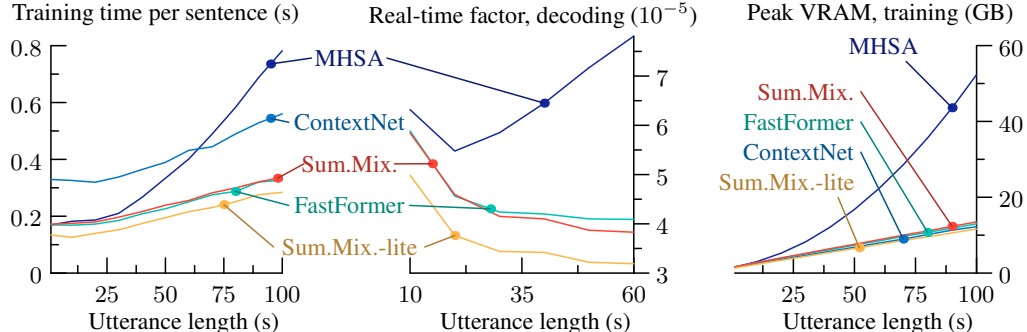

Figure 2: Efficiency measurements and real-time factor analysis. The left- and right-most curves represent the average time as well as the peak VRAM consumption to process a sequence of various lengths during training. The curve in the middle shows the real-time factor for trained ASR systems. The standard Branchformer equipped with MHSA exhibits a quadratic increase in all scenarios while the rest of the models are linear. SummaryMixing-lite is the fastest and cheapest alternative.

for the attention branch. ContextNet definition follows the architecture description described in the original work by Han et al. (2020) corresponding to $\alpha = 1.5$ and leading to 65M parameters. For the RTF analysis, models are pre-trained from our Librispeech experiments from section 3.3.1.

### 3.2.1 RESULTS AND DISCUSSION

Figure 2 depicts the obtained efficiency and RTF measurements for the considered models. It is clear from the training time, RTF as well as peak VRAM consumption that the MHSA-equipped Branchformer leads to a quadratic increase in required resources. For instance, with speech utterances of duration 100 seconds, we see a 2.5 times longer training time compared to the SummaryMixing Branchformer and an explosion of the VRAM consumption from 11.6 GB for the SummaryMixing Branchformer to 52 GB for the MHSA Branchformer. The latter phenomenon is particularly critical as it drastically impacts the price and availability of the required hardware. This is also true for the RTF as the MHSA Branchformer takes twice the amount of time necessary for the SummaryMixing Branchformer to transcribe 60 seconds of speech. This is particularly critical in applications with latency constraints. In short, ASR decoding with a SummaryMixing Branchformer will only be memory-bounded compared to the memory and latency boundaries for an MHSA Branchformer. Indeed, the RTF latency of the SummaryMixing Branchformer does not increase with the length of the audio signal. Also, the memory bound will be rapidly reached by the MHSA as it is quadratic compared to SummaryMixing which is linear, as shown by Figure 2.

However, typical speech utterances from the training sets of academic benchmarks for speech recognition barely reach 30 seconds. Below this mark, the MHSA-equipped Branchformer seems to only underperform at the VRAM level as the difference in training time is reduced compared to the other systems. Yet, it remains slightly slower than all the alternatives except for the one-second-long sentences where only the SummaryMixing-lite Branchformer is faster. The backward pass offers the largest gains as, with a 100-second long utterance, the MHSA Branchformer is 3.1 times slower than its SummaryMixing-lite equivalent while, for the same sequence size, the delta for the inference time is reduced to a 1.4 times faster inference for the SummaryMixing-lite Branchformer. Overall, the introduced SummaryMixing appears to be much cheaper and faster than the MHSA-equipped Branchformer as it eliminates the quadratic complexity. The latter efficiency gains, however, must be validated with real speech recognition and understanding accuracies. Indeed, most efficient alternatives to MHSA for ASR fail to reach the same level of performance (Peng et al., 2022).

### 3.3 SPEECH RECOGNITION AND UNDERSTANDING EXPERIMENTS

This section aims at validating the lack of usefulness of self-attention for ASR and SLU systems.

**Speech recognition tasks details.** Speech recognition with an encoder-decoder architecture based on CTC (Graves & Graves, 2012) combined with Transformer decoding is performed (Karita et al., 2019). Additional experiments with CTC-only training and decoding are also performed to remove entirely self-attention from the ASR architecture. ASR is conducted on five datasets of different languages and complexities in terms of acoustic conditions and available training data: LibriSpeech (960 hours of clean and read English speech) (Panayotov et al., 2015), CommonVoice (*version 13.0*, clean and noisy read speech) (Ardila et al., 2020) Italian (300 hours), Dutch (40 hours), and French (730 hours) as well as AISHELL-1 (170 hours of clean and read Mandarin speech) (Bu et al., 2017) and Ted-Lium 2 (207 hours of recorded Ted talks in English) (Rousseau et al., 2014). Evaluations are conducted on the official sets of each dataset. On Librispeech, models are evaluated without a language model on the *dev-clean* set, and compared with a transformer language model shallow fusion on the *test-clean* and *test-other*. No language models are used for the other datasets. All baselines have been retrained following the official recipes of SpeechBrain (version *0.5.14*) to enable a controlled comparison. The full list of training parameters is available in Appendix A.6.

**Speech understanding tasks details.** Models are compared across two tasks of speech understanding with the SLURP dataset from Bastianelli et al. (2020) for scenarios, actions, and entity classification and the Google speech commands dataset for keyword spotting. SLURP contains around 58 hours of speech material made of 100 participants simulating interactions with a house robot assistant. Participants were asked to voluntarily move in the room while talking, or not to face the microphone to mimic real-world interactions. Google speech commands offers short sentences and labels corresponding to a vocabulary of 35 commands. All experiments strictly follow the SpeechBrain recipes (version *0.5.14*). All models are trained with a single Nvidia RTX 3090.

**Model architectures.** All baselines are selected for ASR with Librispeech as an initial performance and training cost analysis. Then, a reduced subset of the baselines including the Branchformer with MHSA, SummaryMixing, SummaryMixing-lite, and FastFormer is used across the seven other datasets for further comparison. This is done to lower the number of unnecessary training runs. We stick to the original recipe of SpeechBrain to start from a state-of-the-art Branchformer and Conformer, leading to model sizes of roughly 110M parameters for all considered methods except the SummaryMixing Conformer (103M), the FastFormer Branchformer (101M), the ContextNet (100M), the SummaryMixing-lite Branchformer (96M) and the CNN-only Branchformer (77M). Encoders are set up comparably to the efficiency analysis with an internal dimension of 512 with 18 blocks. All models share the same decoder architecture with a dense linear for the CTC predictions as well as a standard Transformer decoder with MHSA and 6 blocks. The vocabulary contains one thousand tokens obtained with BPE. The Transformer language model is pre-trained and obtained from the SpeechBrain toolkit. For the remaining ASR and SLU tasks, and following the reduced set of baselines selected from Librispeech results, we propose to conduct another set of experiments with smaller architectures for a sensitivity analysis. A specific parametrization of the models is conducted leading to 21M parameters for all of them except for SummaryMixing-lite Branchformer (18.5M).

### 3.3.1 SPEECH RECOGNITION ANALYSIS ON LIBRISPEECH

Table 1 lists the word error rates (WERs) as well as the total training time and peak VRAM consumption on Librispeech. All models, including the CNN-only alternatives, achieve competitive recognition rates. For instance, the CNN-only Branchformer achieved a WER of 3.1% on the *dev-clean* set, even beating the standard Transformer with 3.3% WER. This finding supports the evidence that MHSA may not be necessary for the encoder of speech recognizer systems to achieve good accuracies. It is interesting to notice, however, that using MHSA to incorporate the global context slightly improves the overall word error rates while strongly impacting the needed resources. In fact, the 0.2%, 0.2%, and 0.6% improvements on the *dev-clean*, *test-clean* and *test-other* sets respectively of the MHSA Branchformer compared to the CNN-only Branchformer is done at the expense of 49 hours of compute, representing an increase of 58% in training time. The necessary VRAM also goes from 22 GB to 45 GB for the CNN-only and MHSA versions respectively (i.e. increase of 105%).

SummaryMixing Branchformers and Conformer reduce this disproportionate resource impact while preserving or improving the performance. Indeed, the SummaryMixing-lite Branchformer reduces the absolute performance gap to 0.1%, 0.0% and 0.1% on the *dev-clean*, *test-clean* and *test-other* compared to MHSA while only adding 15 hours of training time compared to the Branchformer

Table 1: Speech recognition results on encoder-decoder models with CTC plus Transformer decoding on the Librispeech dataset. "Summary Decoder" adds a SummaryMixing decoder to the SummaryMixing encoder. "Dev-clean" word error rates (WER) are obtained without a language model while "Test-clean" and "Test-other" use shallow fusion and a Transformer language model. "GPU Hours" represent the total training time obtained by multiplying the real-time by the number of GPUs. "VRAM" reports the peak amount of VRAM over the four GPUs during training. Lower is better.

| Encoder | Variant | Dev-clean WER % | Test-clean WER % | Test-other WER % | GPU hours | VRAM GB |
|---------|---------|-----------------|------------------|------------------|-----------|---------|
| ContextNet | N.A. | 3.3 | 2.3 | 5.9 | 160 | 25 |
| Transformer | Self-attention | 3.3 | 2.3 | 5.5 | 129 | 40 |
| Conformer | Self-attention | **2.8** | 2.3 | 5.4 | 137 | 46 |
| Branchformer | Self-attention | 2.9 | 2.2 | **5.1** | 132 | 45 |
|  | CNN Only | 3.1 | 2.4 | 5.7 | **83** | 22 |
|  | HyperMixer | 3.1 | 2.3 | 5.6 | 126 | 30 |
|  | FastFormer | 3.0 | 2.2 | 5.4 | 96 | 23 |
|  | **Proposed** |  |  |  |  |  |
| Conformer | SummaryMixing | **2.8** | **2.1** | **5.1** | 98 | **21** |
| Branchformer | SummaryMixing-lite | 3.0 | 2.2 | 5.2 | 98 | 23 |
|  | SummaryMixing | 2.9 | 2.2 | **5.1** | 105 | 26 |
|  | +Summary Decoder | 3.1 | 2.3 | 5.3 | 104 | 26 |

CNN-only. This is also true for the VRAM consumption which is halved compared to MHSA. Moreover, the SummaryMixing Branchformer closes the gap with MHSA by achieving strictly the same performance with a reduction of the peak VRAM from 45 GB to 26 GB. ContextNet, despite achieving respectable performance, is not at the level initially reported by Han et al. (2020). However, there exists no replication of such results. Finally, the SummaryMixing Conformer also beats the standard Conformer with MHSA and reaches the best *test-clean* and *test-other* WER among all the models while halving the required VRAM from 46 GB for the MHSA variant to 21 GB and exhibiting a 28% faster training time. The architecture of the SummaryMixing Conformer is presented in Appendix A.4.

**Removing self-attention from the ASR attentional decoder.** In practice, we empirically find that replacing cross-attention in the decoder leads to unacceptable performance degradation. This is explained by the fact that cross-attention is critical in capturing the alignment between the text and the audio. However, keeping cross-attention and replacing the remaining self-attention with SummaryMixing leads to competitive performance as shown by the last row of Table 1. Efficiency-wise, the gains are not as impressive as with the encoder part since the sequence length of the decoded text is much shorter than the input audio. We also validate the performance of the SummaryMixing Branchformer encoder without any attention using CTC-only decoding after CTC plus attention training, which may be common deployment or evaluation cases to reduce the decoding latency (Zhang et al., 2022). Without any language model, the Branchformer obtained 2.6% and 6.2% of WER on the *test-clean* and *test-other* sets compared to 2.5% and 6.4% for the SummaryMixing Branchformer. Such numbers are comparable to the latest SOTA systems and even beat well-established dual-decoding models from WeNet (Zhang et al., 2022).

**Removing the attentional decoder.** Finally, we also performed CTC-only training on the two most promising architectures from Librispeech: Conformer with MHSA and SummaryMixing. In this scenario, the decoder is not able to recover from the lack of self-attention in the encoder part. The experimental setups are shown in the Appendix A.6. With CTC greedy decoding, on the *dev-clean*, *test-clean*, and *test-other*, the Conformer with MHSA (28.8M) achieves 3.5%, 3.7%, 9.2% WERs respectively while the SummaryMixing-Conformer (26.5M) reaches 3.5%, 3.7%, and 9.4% WERs. Even without MHSA in the architecture, SummaryMixing performs on par with MHSA.

Table 2: Summary of the speech recognition, keyword spotting and speech understanding results. ASR accuracy is expressed in word error rate (WER, lower is better) for the test sets of CommonVoice Dutch (*"Nl."*, 40 hours), Italian (*"It."*, 300 hours), French (*"Fr."*, 730 hours), AISHELL-1 denoted as *"AI."* (170 hours) and Ted-Lium 2 denoted as *"Ted."* (207 hours). No language model is applied. SLU results are expressed in SLU-F1 for SLURP and accuracy for Google Speech Command (GSC).

| Metric
Model size | WER ↓
19–21M | | | | | WER ↓
90–100M | | | | | F1 ↑
60M | Acc. ↑
10M |
|---|---|---|---|---|---|---|---|---|---|---|---|---|
| Encoder | Nl. | It. | Fr. | AI. | Ted. | Nl. | It. | Fr. | AI. | Ted. | SLURP | GSC |
| Branchformer | 33.4 | 12.7 | 14.5 | 6.2 | 11.3 | 32.6 | 10.5 | 11.0 | **5.7** | 7.9 | 0.771 | 98.06 |
| —*FastFormer* | 33.2 | 13.0 | **13.4** | 8.1 | 10.2 | 33.9 | 10.9 | 10.9 | 6.1 | 8.5 | — | — |
| —*SummaryMixing* | 33.2 | **12.1** | **13.4** | **6.1** | **9.9** | **31.5** | **10.4** | **10.8** | **5.7** | **7.8** | **0.773** | **98.16** |
| —*SummaryMixing-lite* | **33.1** | 12.5 | 14.1 | 6.3 | **9.9** | 32.0 | 10.5 | 11.3 | 5.9 | 8.1 | 0.770 | 98.06 |

### 3.3.2 EXTENDED SPEECH RECOGNITION AND UNDERSTANDING EXPERIMENTS

In this set of experiments, we focused on the Branchformer with MHSA or FastFormer versus SummaryMixing Branchformer comparison by extending the number of languages and acoustic conditions for ASR. We also introduce two more tasks including keyword spotting and SLU. We selected the Branchformer instead of the conformer as it obtained better performance on average on our Librispeech experiment and in the literature (Kim et al., 2023). All the results are reported in Table 2. For ASR, all datasets are much harder than LibriSpeech, due to important variations in the acoustic conditions but also in the amount of available data.

Focusing on ASR, it is worth noting that the SummaryMixing Branchformers outperform the MHSA Branchformer in terms of WER with all datasets, and hence in all data regimes. Indeed, on average over all the model sizes and datasets, the SummaryMixing and SummaryMixing-lite Branchformers reduced the WER by 0.5% and 0.2% absolute compared to the MHSA Branchformer. Such results validate the hypothesis that MHSA is not strictly necessary to achieve state-of-the-art speech recognition performance even across a wide range of acoustic conditions and languages.

The previous conclusion also extends to other tasks as the SummaryMixing Branchformers are able to reach their MHSA counterpart on spoken language understanding with an SLU-F1 score of 0.773 and 0.771 for the SummaryMixing and SummaryMixing-lite compared to 0.771 for MHSA. For keyword spotting, SummaryMixing-lite reaches strictly the same accuracy as MHSA, while SummaryMixing improves it by 0.1% absolute. These results indicate that MHSA may also not be strictly necessary for other speech-related tasks and that it can be replaced with SummaryMixing to save training time, energy, and memory without sacrificing on performance.

## 4 CONCLUSION

This work proposed SummaryMixing, a novel linear-time complexity block removing the need for self-attention in speech recognition and understanding encoders. SummaryMixing is based on the following assumptions: (a) the acoustic modeling component of speech systems does not require multi-head self-attention; and (b) an efficient and cheaper global context vector taking the form of a summary of each speech utterance is sufficient to reach top-of-the-line speech recognition and understanding. SummaryMixing-equipped Conformers and Branchformers outperform state-of-the-art MHSA-equipped equivalent architectures while exhibiting a linear complexity, leading to a reduction of up to 28% in training time as well as more than half of the original VRAM consumption. SummaryMixing also leads to significantly faster inference and decoding times for offline speech recognition and understanding and could be extended to any speech encoder.

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
