# A  Supplementary Material

The supplementary material provides different resources complementing the main description and evaluation of SummaryMixing:

1. Appendix A.1 details the relationship between HyperMixing and SummaryMixing.
2. Appendix A.2 details an engineering enhancement associated with SummaryMixing to reduce its number of parameters.
3. Appendix A.5 reports results showing the ASR accuracy of MHSA and SummaryMixing as a function of the audio length.
4. Appendix A.3 describes how SummaryMixing-lite is integrated to the Branchformer.
5. Appendix A.4 describes how SummaryMixing is integrated to the Conformer.
6. Appendix A.6 list the hyperparameters of all conducted ASR experiments.

## A.1  Relationship between SummaryMixing and the HyperMixer

The analysis of the relationship between SummaryMixing and the HyperMixer (Mai et al., 2023) is relegated to the appendix due to its length. The following will first follow the original presentation, and then re-write the mathematics to relate them to SummaryMixing.

The HyperMixer starts from MLP Mixer (Tolstikhin et al., 2021), which mixes "tokens", here feature vectors $\mathbf{x}_t$. The way these feature vectors are mixed is dimension by dimension. Denoted with $\mathbf{x}_i^T$ feature dimension $i$ across time. The output of MLP Mixer, again, for a single dimension, is

$$\mathbf{h}_i^T = \mathrm{MLP}(\mathbf{x}_i^T) = \mathbf{W}_1 \cdot \sigma(\mathbf{W}_2^T \cdot \mathbf{x}_i^T), \tag{2}$$

where $\sigma()$ is a nonlinearity, and $\mathbf{W}_i \in \mathbb{R}^{T \times D'}$ are weight matrices. The first dimensions of both weight matrices are the length of the input. If the weight matrices are trained directly, the input must therefore be of fixed length which is not common in speech processing.

The HyperMixer (Mai et al., 2023) therefore makes both weight matrices $\mathbf{W}_k$ variable-height, by making them functions of the input. Each row of $\mathbf{W}_k$ is a function of the corresponding per-time feature vector $\mathbf{x}_t$:

$$\mathbf{W}_k(\mathbf{X}) = \begin{bmatrix} \mathrm{MLP}_k(\mathbf{x}_0) \\ \vdots \\ \mathrm{MLP}_k(\mathbf{x}_T) \end{bmatrix}. \tag{3}$$

It is also possible to add a positional encoding to $\mathbf{x}_t$.

The reason for Mai et al. (2023) to use the word "HyperMixer" is the analysis as an MLP Mixer with the parameters not chosen directly, but by a "hyper-network". This is an unusual use of the term "hyper", since hyper-networks are immediately dependent on the input.

Up to this point, the presentation of Mai et al. (2023) has been followed with only notational changes, but from now the HyperMixer will be analysed differently. First, a key question is why the HyperMixer is faster. Mai et al. (2023) cite "simplicity" as the key to their performance improvements, which is imprecise. The answer is linear time complexity, which is not obvious from the presentation so far. To express the output of the HyperMixer per element of the matrix $\mathbf{H}$, rewrite (2), using $[\cdot]_{ij}$ to denote element $(i, j)$ of a matrix:

$$[\mathbf{H}]_{t,i} = \sum_{j=1}^{D'} [\mathbf{W}_1]_{t,j} \cdot \left[\sigma(\mathbf{W}_2^T \cdot \mathbf{X})\right]_{j,i}. \tag{4}$$

Now, this can be reformulated per time step by fixing $t$ and recognising the expression as a vector-matrix product, where the vector is given by (3):

$$\mathbf{h}_t = [\mathbf{W}_1]_t \cdot \sigma(\mathbf{W}_2^T \cdot \mathbf{X}) = \mathrm{MLP}_1(\mathbf{x}_t) \cdot \sigma(\mathbf{W}_2^T \cdot \mathbf{X}). \tag{5}$$

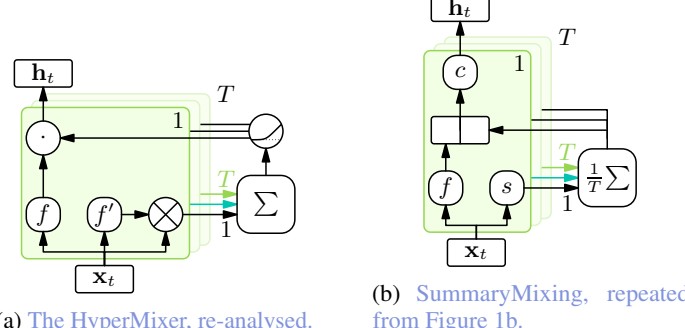

(a) The HyperMixer, re-analysed.

(b) SummaryMixing, repeated from Figure 1b.

Figure 3: Comparison between the HyperMixer and SummaryMixing.

It has become clear that the HyperMixer performs a per-time step transformation of $\mathbf{x}_t$, and then a linear transformation by $\sigma(\mathbf{W}_2^T \cdot \mathbf{X})$. $\sigma(\mathbf{W}_2^T \cdot \mathbf{X})$ does not have $t$ in it and therefore must be a global projection matrix.

The elements of the global projection matrix are

$$\left[\sigma(\mathbf{W}_2^T \cdot \mathbf{X})\right]_{j,i} = \sigma\left(\sum_{t'=1}^{T} [\mathbf{W}_2]_{t',j} \cdot [\mathbf{X}]_{t',i}\right). \tag{6}$$

Note that $\sigma(\cdot)$ is a per-element nonlinearity. To make this expression's dependency on the length $T$ of the input clear, this can be written more simply as the sum of a per-time cross product:

$$\sigma(\mathbf{W}_2^T \cdot \mathbf{X}) = \sigma\left(\sum_{t'=1}^{T} \mathrm{MLP}_2(\mathbf{x}_{t'}) \times \mathbf{x}_{t'}\right). \tag{7}$$

Our re-analysis of the HyperMixer is shown in Figure 3a. To keep to linear complexity in the length of the input, most operations are per time step. There is a local transformation of the input ($f$ in the figure, or $\mathrm{MLP}_1$ in the original description). Separately, there is a per-time step contribution to a global sum, which is given by $f'(\mathbf{x}_{t'}) \times \mathbf{x}_{t'}$ (where $f'$ is written $\mathrm{MLP}_2$ in the original description). The size per-time step contribution, crucially, is independent of the length of the input. The result of the global sum is taken through nonlinearity $\sigma$ and used as a projection matrix for each of the local transformations of the input.

A comparison between the re-analysed HyperMixer in Figure 3a and SummaryMixing in Figure 3b shows a similar structure. However, in the HyperMixer, only one part of the local contribution, function $f'$, is trainable, and the combination of local and global information is fixed: a global projection applied to a local vector. On the other hand, in SummaryMixing, the local contribution is a completely trainable function $s$, an average is taken instead of a sum (though layer normalisation in the HyperMixer may have led to the same effect), and the combination of local and global information, again, a trainable function, after concatenation.

## A.2 SUMMARYMIXING WITH INPUT CHUNKING

In the context of SummaryMixing, a simple trick can be applied to the different transformations to reduce significantly the number of parameters without affecting the size of the hidden dimensions. We refer to this trick as "input chunking". The core idea is that each input tensor can be divided into $n$ chunks along the feature dimension (i.e. last dimension) and be processed independently by smaller neural networks instead of a larger one attending to the full feature dimension. The latter creates $n$ smaller neural networks that will be specialized in always dealing with the same chunk of the input tensor.

Such a process can be formally described as follows. Let $\mathbf{x}_t$ be the input tensor of dimension $[B, T, D]$ with $B$ the batch size, $T$ the number of time steps and $D$ the hidden or feature dimension. The $D$ dimension of $\mathbf{x}_t$ can be divided into $n$ chunks to reduce the size of the $s$ and $f$ functions

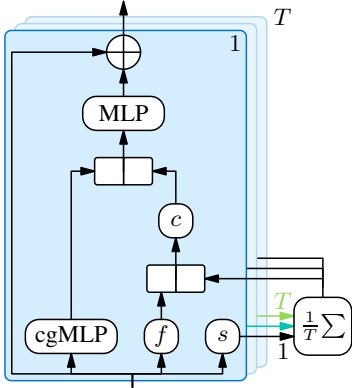

Figure 4: Branchformer equipped with SummaryMixing. The cgMLP branch provides local information while the SummaryMixing branch gives global information.

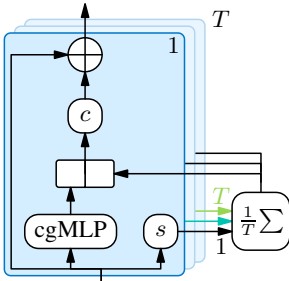

Figure 5: Branchformer equipped with SummaryMixing-lite. The cgMLP branch also acts as a transformation function ($f$ in Figure 4) for the SummaryMixing operation.

that only need to be created $n$ times. In practice, as $s$ and $f$ are dense non-linear neural networks, we will create $n$ versions of them, but with $n$-times reduced input and output dimensions. The $n$ different outputs of $s$ and $f$ can then be concatenated to reproduce the original output dimension. In our SummaryMixing, the $n$ summary and transformation functions are untied, i.e., have different weight parameters, to further increase the modeling capacities of the model. Therefore, the model ends up with $n$ different linear layers for the $s$ and $f$ functions. This helps to reduce the number of neural parameters as $\frac{D}{n} \times \frac{D}{n} \times n \leq D \times D$. For instance, the number of parameters goes from 1.2M to 262k for a layer of 1024 neurons and four chunks.

## A.3    BRANCHFORMER WITH SUMMARYMIXING AND SUMMARYMIXING-LITE

Figure 4 shows a detailed illustration for the architecture of a Branchformer layer equipped with SummaryMixing. The inputs of each layer go to both the cgMLP branch and the SummaryMixing branch ($f$, $s$, and $c$ in Figure 4). In addition, since both the cgMLP branch and the Transformation function $f$ in the SummaryMixing branch extract local information, we also propose to merge $f$ with cgMLP. As shown by Figure 5, this merge makes the SummaryMixing operation fully integrated into the Branchformer architecture, leading to a SummaryMixing-lite structure which has even less complexity in terms of neural parameters compared to SummaryMixing.

## A.4    CONFORMER WITH SUMMARYMIXING

Figure 6 shows the architecture of a Conformer layer. Between the two "macaron-like" MLP modules is an self-attention or SummaryMixing module for the global information and a convolutional module for the local information. The main design differences of Conformer and Branchformer is that Conformer processes global and local information in a sequential way while the latter processes global and local information in parallel. Our proposed SummaryMixing Conformer replaces the self-attention module with a SummaryMixing module in each Conformer layer.

## A.5    AUDIO DURATION SENSITIVITY ANALYSIS

The sensitivity of SummaryMixing, Fastformer, and self-attention to the variation of the duration of audio files during speech recognition decoding is investigated in this section. In particular, this experiment aims to ensure that the removal of self-attention does not harm the performance of the ASR model for longer sentences. To achieve this, we evaluate the WER of the small Branchformers trained on the Tedlium 2 dataset and presented in Table 2 on ten sets of sentences of increasing duration. As a reminder, this ASR model is a Branchformer encoder with a transformer decoder trained jointly with CTC and without any language model. These sets are designed by taking the test set of Tedlium and splitting it into 10 partitions where sentences fall into buckets of corresponding lengths. We

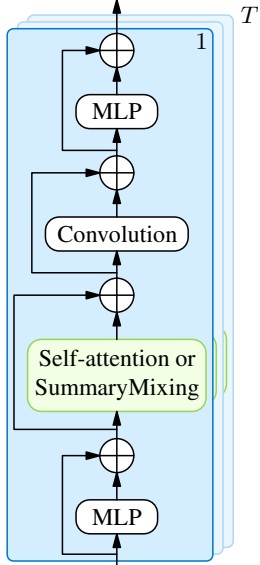

Figure 6: The Conformer. It uses an attention or SummaryMixing module for the global information and a convolutional module for the local information.

then compute the WER of the Branchformers equipped with SummaryMixing, SummaryMixing-lite, Fastformer, and self-attention and report the result for each bucket of increasing duration in Figure **??**. From the results, it is clear that not only both SummaryMixing and SummaryMixing-lite perform the best, but also that longer sentences do not harm SummaryMixing more than MHSA. It appears to be the opposite as the WER increases more rapidly for MHSA than SummaryMixing with the increase in audio duration. Hence, we can conclude that SummaryMixing does not alter the long-term context learning capabilities of encoder-decoder ASR systems when replacing MHSA.

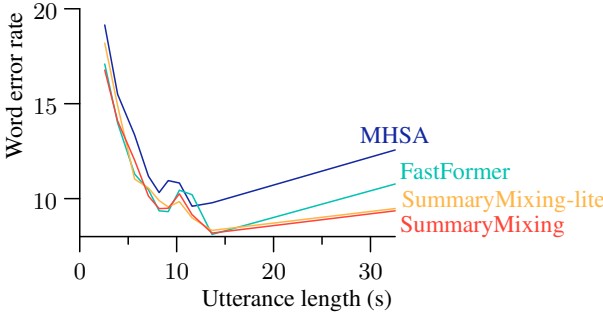

Figure 7: Evolution of the WER of different Branchformers encoder-decoder (+ CTC) ASR systems trained on Tedlium 2 and tested on 10 sets of sentences of increasing duration coming from the Tedlium 2 test set. The attention cell of the Branchformer encoder can either be Multi-head self-attention, SummaryMixing, SummaryMixing-lite or Fastformer. SummaryMixing is not more impacted by longer sentences than MHSA.

## A.6 SPEECH RECOGNITION DETAILS

The following tables describe the precise set of hyperparameters used for the newly introduced models for ASR experiments on the Librispeech, CommonVoice, Tedlium, AISHELL-1, and AMI datasets. The parameters of the models already available in SpeechBrain are omitted as the hyperparameter files can be found with SpeechBrain *v0.5.14*.

Table 3: Hyperparameters and architecture details for the ASR experiments conducted on Librispeech with joint CTC/Attention training.

| Parameter | Branchformer *SummaryMixing* | Branchformer *SummaryMixing-lite* | Conformer *SummaryMixing* |
|---|---|---|---|
| **Optimization** | | | |
| Epochs | 120 | 120 | 120 |
| GPUs | 4 | 4 | 4 |
| Batching | Dynamic | Dynamic | Dynamic |
| Batch Len. | 500s | 500s | 500s |
| Optimizer | AdamW | AdamW | AdamW |
| LR Scheduler | noAM | no AM | no AM |
| Max. LR | 5e-3 | 5e-3 | 5e-3 |
| Warmup steps | 30k | 30k | 30k |
| Weight Decay | 0.001 | 0.001 | 0.001 |
| CTC weight | 0.3 | 0.3 | 0.3 |
| Attention weight | 0.7 | 0.7 | 0.7 |
| **Augmentations** | | | |
| SpecAugment | True | True | True |
| Time warp window | 5 | 5 | 5 |
| Freq. Masks | 2 | 2 | 2 |
| Masks width | 30 | 30 | 30 |
| Time masks | 3 | 3 | 3 |
| Masks width | 40 | 40 | 40 |
| Speed Perturb. | True | True | True |
| Speeds | [95,100,105] | [95,100,105] | [95,100,105] |
| **CNN FrontEnd** | | | |
| Input | 80 FBanks | 80 FBanks | 80 FBanks |
| Type | Conv1D | Conv1D | Conv1D |
| Layers | 2 | 2 | 2 |
| Filters | (64,32) | (64,32) | (64,32) |
| Kernel Size | (3,3) | (3,3) | (3,3) |
| Strides | (2,2) | (2,2) | (2,2) |
| **Encoders** | | | |
| Model dim. | 512 | 512 | 512 |
| Heads | 4 | 4 | 4 |
| Blocks | 18 | 18 | 18 |
| Dropout | 0.1 | 0.1 | 0.1 |
| Activation | GeLU | GeLU | GeLU |
| Attention | SummaryMixing | SummaryMixing-lite | RelPosMHAXL |
| cgMLP Lin. | 3072 | 3072 | 3072 |
| cgMLP Kernel | 31 | 31 | 31 |
| Strides | (2,2) | (2,2) | (2,2) |
| **Decoders** | | | |
| Model dim. | 512 | 512 | 512 |
| Type | Transformer | Transformer | Transformer |
| CTC | True | True | True |
| Inp. chunk./Heads | 4 | 4 | 4 |
| Blocks | 6 | 6 | 6 |
| Dropout | 0.1 | 0.1 | 0.1 |
| Activation | GeLU | GeLU | GeLU |
| Vocabulary type | BPE | BPE | BPE |
| Vocabulary size | 5000 | 5000 | 5000 |
| **Decoding (Transformer LM)** | | | |
| beam size | 66 | 66 | 66 |
| LM Weight | 0.6 | 0.6 | 0.6 |
| Temperature | 1.15 | 1.15 | 1.15 |

Table 4: Hyperparameters and architecture details for the ASR experiments conducted on Librispeech with CTC only training.

| Parameter | Conformer Self-attention | Conformer SummaryMixing |
|---|---|---|
| **Optimization** | | |
| Epochs | 500 | 500 |
| GPUs | 4 | 4 |
| Batching | Dynamic | Dynamic |
| Batch Len. | 1700s | 1700s |
| Optimizer | AdamW | AdamW |
| LR Scheduler | noAM | no AM |
| Max. LR | 1e-3 | 1e-3 |
| Warmup steps | 7.5k | 7.5k |
| Steps of Keeping Max. LR | 43k | 43 K |
| Weight Decay | 5e-4 | 5e-4 |
| **Augmentations** | | |
| SpecAugment | True | True |
| Time warp window | 5 | 5 |
| Freq. Masks | 2 | 2 |
| Masks width | 27 | 27 |
| Time masks | 7 | 8 |
| Masks width | 5e-2 $\times$ Utt. Len. | 5e-2 $\times$ Utt. Len. |
| Speed Perturb. | True | True |
| Speeds | [95,100,105] | [95,100,105] |
| **CNN FrontEnd** | | |
| Input | 80 FBanks | 80 FBanks |
| Type | Conv1D | Conv1D |
| Layers | 2 | 2 |
| Filters | (64,32) | (64,32) |
| Kernel Size | (3,3) | (3,3) |
| Strides | (2,2) | (2,2) |
| **Encoders** | | |
| Model dim. | 256 | 256 |
| Inp. chunk./Heads | 4 | 4 |
| Feedforward dim. | 1024 | 1024 |
| Blocks | 18 | 18 |
| Dropout | 0.1 | 0.1 |
| Activation | GeLU | GeLU |
| Attention | RelPosMHAXL | SummaryMixing |
| Conv. Module Kernel | 31 | 31 |
| **Decoders** | | |
| Type | CTC-only | CTC-only |
| Vocabulary type | BPE | BPE |
| Vocabulary size | 128 | 128 |
| **Decoding (Greedy CTC decoding)** | | |
| beam size | 1 | 1 |

Table 5: Hyperparameters and architecture details for the ASR experiments conducted on Common-Voice 13.0 with joint CTC/Attention training. No language model is applied. The differences between languages are mentioned in the "optimization" column.

| Parameter | Branchformer
*SummaryMixing* | Branchformer
*SummaryMixing-lite* | Branchformer
*Fastformer* |
|---|---|---|---|
| **Optimization** | | | |
| Epochs (nl/it/fr) | 120/100/100 | 120/100/100 | 120/100/100 |
| GPUs | 2 | 2 | 2 |
| Batching | Dynamic | Dynamic | Dynamic |
| Batch Len. | 400s | 400s | 400s |
| Optimizer | AdamW | AdamW | AdamW |
| LR Scheduler | no AM | no AM | no AM |
| Max. LR | 5e-3 | 5e-3 | 5e-3 |
| Warmup steps (nl/it/fr) | 10k/10k/25k | 10k/10k/25k | 10k/10k/25k |
| Weight Decay | 0.001 | 0.001 | 0.001 |
| CTC weight | 0.3 | 0.3 | 0.3 |
| Attention weight | 0.7 | 0.7 | 0.7 |
| **Augmentations** | | | |
| SpecAugment | True | True | True |
| Time warp window | 5 | 5 | 5 |
| Time warp mode | bicubic | bicubic | bicubic |
| Freq. Masks | 2 | 2 | 2 |
| Masks width | 30 | 30 | 30 |
| Time masks | 3 | 3 | 3 |
| Masks width | 40 | 40 | 40 |
| **CNN FrontEnd** | | | |
| Input | 80 FBanks | 80 FBanks | 80 FBanks |
| Type | Conv1D | Conv1D | Conv1D |
| Layers | 2 | 2 | 2 |
| Filters | (64,32) | (64,32) | (64,32) |
| Kernel Size | (3,3) | (3,3) | (3,3) |
| Strides | (2,2) | (2,2) | (2,2) |
| **Encoders** | | | |
| Model dim. (large,small) | (512,256) | (512,256) | (512,256) |
| Inp. chunk./Heads | 4 | 4 | 4 |
| Blocks (large, small) | (18,12) | (18,12) | (18,12) |
| Dropout | 0.1 | 0.1 | 0.1 |
| Activation | GeLU | GeLU | GeLU |
| Attention | SummaryMixing | SummaryMixing-lite | Fastformer |
| cgMLP Lin. (large, small) | (3072,1536) | (3072,1536) | (3072,1536) |
| cgMLP Kernel | 31 | 31 | 31 |
| **Decoders** | | | |
| Model dim. | 256 | 256 | 256 |
| Type | Transformer | Transformer | Transformer |
| CTC | True | True | True |
| Heads | 4 | 4 | 4 |
| Blocks (large, small) | (6,4) | (6,4) | (6,4) |
| Dropout | 0.1 | 0.1 | 0.1 |
| Activation | GeLU | GeLU | GeLU |
| Vocabulary type | BPE | BPE | BPE |
| Vocabulary size (nl/it/fr) | 350 | 1000 | 1000 |
| **Decoding (No LM)** | | | |
| beam size | 10 | 10 | 10 |
| LM Weight | 0.6 | 0.6 | 0.6 |
| Temperature | 1.15 | 1.15 | 1.15 |

Table 6: Hyperparameters and architecture details for the ASR experiments conducted on AISHELL-1 with joint CTC/Attention training. No language model is applied. This recipe is trained with a two-stage optimisation process. AdamW is used first and then the model is fine-tuned with SGD.

| Parameter | Branchformer *SummaryMixing* | Branchformer *SummaryMixing-lite* | Branchformer *Fastformer* |
|---|---|---|---|
| **Optimization** | | | |
| Epochs (large, small) | (120, 360) | (120, 360) | (120, 360) |
| GPUs | 2 | 2 | 2 |
| Batching | Dynamic | Dynamic | Dynamic |
| Batch Len. | 300s | 300s | 300s |
| Optimizer one | AdamW | AdamW | AdamW |
| Optimizer two | SGD | SGD | SGD |
| LR Scheduler | no AM | no AM | no AM |
| Max. LR one (large, small) | 8e-3 | 8e-3 | (8e-3, 8e-4) |
| LR two (large, small) | 2e-5 | 2e-5 | (2e-5, 2e-4) |
| Warmup steps | 25k | 25k | 25k |
| Weight Decay | 0.01 | 0.01 | 0.01 |
| CTC weight | 0.3 | 0.3 | 0.3 |
| Attention weight | 0.7 | 0.7 | 0.7 |
| **Augmentations** | | | |
| SpecAugment | True | True | True |
| Time warp window | 5 | 5 | 5 |
| Time warp mode | bicubic | bicubic | bicubic |
| Freq. Masks | 2 | 2 | 2 |
| Masks width | 30 | 30 | 30 |
| Time masks | 2 | 2 | 2 |
| Masks width | 40 | 40 | 40 |
| **CNN FrontEnd** | | | |
| Input | 80 FBanks | 80 FBanks | 80 FBanks |
| Type | Conv1D | Conv1D | Conv1D |
| Layers | 2 | 2 | 2 |
| Filters | (64,32) | (64,32) | (64,32) |
| Kernel Size | (3,3) | (3,3) | (3,3) |
| Strides | (2,2) | (2,2) | (2,2) |
| **Encoders** | | | |
| Model dim. (large,small) | (512,256) | (512,256) | (512,256) |
| Heads | 4 | 4 | 4 |
| Blocks (large, small) | (18,12) | (18,12) | (18,12) |
| Dropout | 0.1 | 0.1 | 0.1 |
| Activation | GeLU | GeLU | GeLU |
| Attention | SummaryMixing | SummaryMixing-lite | Fastformer |
| cgMLP Lin. (large, small) | (3072,1536) | (3072,1536) | (3072,1536) |
| cgMLP Kernel | 31 | 31 | 31 |
| **Decoders** | | | |
| Model dim. | 256 | 256 | 256 |
| Type | Transformer | Transformer | Transformer |
| CTC | True | True | True |
| Heads | 4 | 4 | 4 |
| Blocks (large, small) | (6,4) | (6,4) | (6,4) |
| Dropout | 0.1 | 0.1 | 0.1 |
| Activation | GeLU | GeLU | GeLU |
| Vocabulary type | BPE | BPE | BPE |
| Vocabulary size | 5000 | 5000 | 5000 |
| **Decoding (No LM)** | | | |
| beam size | 10 | 10 | 10 |

Table 7: Hyperparameters and architecture details for the ASR experiments conducted on Tedlium with joint CTC/Attention training. No language model is applied.

| Parameter | Branchformer *SummaryMixing* | Branchformer *SummaryMixing-lite* | Branchformer *Fastformer* |
|---|---|---|---|
| **Optimization** | | | |
| epochs | 120 | 120 | 120 |
| GPUs (large,small) | 2/4 | 2/4 | 2/4 |
| Batching | Dynamic | Dynamic | Dynamic |
| Batch Len. (large,small) | 800s/400s | 800s/400s | 800s/400s |
| Optimizer | AdamW | AdamW | AdamW |
| LR Scheduler | no AM | no AM | no AM |
| Max. LR | 5e-4 | 5e-4 | 5e-4 |
| Warmup steps (large,small) | 30k/15k | 30k/15k | 30k/15k |
| Weight Decay (large,small) | 5e-2/5e-6 | 5e-2/5e-6 | 5e-2/5e-6 |
| CTC weight | 0.3 | 0.3 | 0.3 |
| Attention weight | 0.7 | 0.7 | 0.7 |
| **Augmentations** | | | |
| SpecAugment | True | True | True |
| Time warp window | 5 | 5 | 5 |
| Time warp mode | bicubic | bicubic | bicubic |
| Freq. Masks | 2 | 2 | 2 |
| Masks width | 30 | 30 | 30 |
| Time masks (large,small) | 7/5 | 7/5 | 7/5 |
| Masks width | 5e-2 × Utt. Len. | 5e-2 × Utt. Len. | 5e-2 × Utt. Len. |
| **CNN FrontEnd** | | | |
| Input | 80 FBanks | 80 FBanks | 80 FBanks |
| Type | Conv1D | Conv1D | Conv1D |
| Layers | 2 | 2 | 2 |
| Filters | (64,32) | (64,32) | (64,32) |
| Kernel Size | (3,3) | (3,3) | (3,3) |
| Strides | (2,2) | (2,2) | (2,2) |
| **Encoders** | | | |
| Model dim. (large,small) | (512,256) | (512,256) | (512,256) |
| Heads | 4 | 4 | 4 |
| Blocks (large, small) | (18,12) | (18,12) | (18,12) |
| Dropout | 0.1 | 0.1 | 0.1 |
| Activation | GeLU | GeLU | GeLU |
| Attention | SummaryMixing | SummaryMixing-lite | Fastformer |
| cgMLP Lin. (large, small) | (3072,1536) | (3072,1536) | (3072,1536) |
| cgMLP Kernel | 31 | 31 | 31 |
| **Decoders** | | | |
| Model dim. | 256 | 256 | 256 |
| Type | Transformer | Transformer | Transformer |
| CTC | True | True | True |
| Heads | 4 | 4 | 4 |
| Blocks (large, small) | (6,4) | (6,4) | (6,4) |
| Dropout | 0.1 | 0.1 | 0.1 |
| Activation | GeLU | GeLU | GeLU |
| Vocabulary type | BPE | BPE | BPE |
| Vocabulary size | 500 | 500 | 500 |
| **Decoding (No LM)** | | | |
| beam size | 20 | 20 | 20 |