# OpenReview forum: "SummaryMixing: A Linear-Complexity Alternative to Self-Attention for Speech Recognition and Understanding"
_ICLR.cc/2024/Conference — Submitted to ICLR 2024_

### Official Review · Reviewer_gqu7 · 2023-10-23

**Soundness:** 3 good
**Presentation:** 3 good
**Contribution:** 2 fair
**Rating:** 5
**Confidence:** 4

**Summary:**

This paper proposes a linear-complexity based architecture as a replacement of self-attention for speech recognition to improve the efficiency. The proposed block has a local branch and a global branch to take both information into account. The local branch is based on a MLP, and the global branch has a MLP and then an average pooling among all the fram

**Strengths:**

* Paper fits ICLR scope well.
* The idea of reducing model complexity through replacing self-attention with linear layers is interesting.
* The proposed method is solid.

**Weaknesses:**

* Presentations need improvement. See question sections for details.
* Literature reviews on ASR are very limited. Please have an individual paragraph in Section 1.1 to describe those.
* Novelty is a concern. The proposed work is an incremental work on the top of BranchFormer and HyperMixer. Although there is a paragraph describing the relationship to HyperMixer, the difference seems to be more on the choice of linear functions. Please explain more on this.
* The improvements from the proposed method seems to depend on the use of a transformer decoder (with self-attention). Need more experiments to verify the parity of the proposed method to self-attention in other conditions. See question sections for details.

**Questions:**

Abstract: “However, attention layers in trained speech recognizers tend to not capture fine-grained pair-wise information.” Please explain what “fine-grained pair-wise information” is and why it is important (although they are explained in the following sections, abstract should be self-explanatory).

Figure 1: The diagrams are a bit confusing. I understand that “T” refers to the total number of frames. However, audiences may likely think of them as different attention heads. Please consider a better way for the diagrams.

Section 2.1 - Multi-head SummaryMixing. Please better explain the heads in SummaryMixing. In self-attention, heads are operating in parallel with different sets of parameters. However, here heads help to reduce the number of parameters. Please consider adding a diagram on this if it helps. In addition, by dividing x_t into n chunks, is it similar to chuck-wise attention or attentions with a limited context window? If that’s the case, please avoid using “head” here.

Section 3.3.1: The results shown in this section have a transformer decoder of 6-blocks-MHSA paired with each system. In this setup, the decoder will capture the global context, even if the encoder doesn’t. However, we can only conclude that self-attention is redundant when self-attention based decoders are used.
The authors also conducted experiments without a transformer decoder (i.e., CTC only) on Branchformer in the supplemental materials. However, there are two more problems to figure out. 1. How does vanilla Conformer/Transformer with CTC performance in this setup? They are the most common architectures for productions than Branchformer. 2. These results are obtained with LM shallow fusion, which can be expensive for deployment. How would the proposed approach perform without shallow fusion? If the language information is needed, please also consider RNN-T decoder.

---

> ### Author Response · Authors · 2023-11-16
> **Thank you 1/2.**
>
> First, we wish to thank the reviewer for mentioning that SummaryMixing is a solid method that fits well the scope of ICLR. We wish to mention that we can provide an anonymized version to the reviewer if necessary. Questions, remarks, and concerns from the reviewer are addressed in the next paragraphs.
>
>
> **Weaknesses**
>
> *Presentations need improvement. See question sections for details.*
>
> Thanks to the reviewer for helping clarify a few aspects of the manuscript. The detailed answers are given in the question section and the paper has been updated accordingly.
>
> *Literature reviews on ASR are very limited. Please have an individual paragraph in Section 1.1 to describe those.*
>
> As suggested, we added a paragraph in Section 1.1 to summarize the ASR domain and how it relates to this work. We invite the reviewer to have another look at this Section.
>
> *Novelty is a concern. The proposed work is an incremental work on the top of BranchFormer and HyperMixer. Although there is a paragraph describing the relationship to HyperMixer, the difference seems to be more on the choice of linear functions. Please explain more on this.*
>
> First of all, we modified Section 2.2 to clarify the use of SummaryMixing both in Conformer and Branchformer as well as Section 2 and the Appendix to clarify the link with HyperMixer. Indeed, we also apply SummaryMixing with Conformer and achieve SOTA, but this was not clearly expressed in the first version of the manuscript. Now, if we see things from the Branchformer point of view, we agree with the reviewer, in proportion, the Sumformer implies only a few changes (and this is a strength from our point of view). However, we would like to first highlight that SummaryMixing is also applied to the Conformer architecture i.e. (not only Branchformer) as shown by Table 1 and the CTC-only experiments. Conformer with SummaryMixing beats the MHSA Conformer as well. As a more general answer, SummaryMixing can replace MHSA in any speech encoder.  Indeed, and as mentioned by the reviewer in the strength of this manuscript, SummaryMixing is about “replacing self-attention with linear layers [...]” not only for Branchformer. In this article we demonstrate that this is feasible with the two most used architectures for ASR: Conformer and Branchformer. Finally, we wish to highlight the fact that Branchformer was accepted at ICML while its contribution was to make two blocks of layers parallel instead of sequential – from our very own point of view, this does not represent much more novelty than SummaryMixing (quantitatively speaking), yet it was accepted and is now used by the community thanks to the easy open-access to the code, which is also true for SummaryMixing.
>
> *The improvements from the proposed method seems to depend on the use of a transformer decoder (with self-attention). Need more experiments to verify the parity of the proposed method to self-attention in other conditions. See question sections for details.*
>
> We thank the reviewer for raising this excellent remark. We invite the reviewer to have a look at the new results added in the top comment of the reviews as well as the last paragrap of Section 3.3.1 We decided to train ASR systems with CTC only, i.e. without any attentional decoder, and reached basically the same performance between MHSA and SummaryMixing on dev-clean and tes-clean. Therefore, we can say that, in the specific context of this experimental protocol, MHSA is not needed to achieve SOTA ASR performance and that SummaryMixing can do just as well WER-wise while drastically speeding up the training and decreasing the VRAM consumption.

---

> > ### Author Response · Authors · 2023-11-16
> > **Thank you 2/2.**
> >
> > **Questions**
> >
> > *Abstract: “However, attention layers in trained speech recognizers tend to not capture fine-grained pair-wise information.” Please explain what “fine-grained pair-wise information” is and why it is important (although they are explained in the following sections, abstract should be self-explanatory).*
> >
> > We agree with the reviewer that this sentence is actually misleading and not adding sufficient context or value to justify its presence in the Abstract. Therefore, we decided to remove it from the abstract and instead rely on the proper explanation of this concept that appears later in the paper, as stated by the review. We kindly ask the reviewer to tell us if any further clarifications should be added.
> >
> > *Figure 1: The diagrams are a bit confusing. I understand that “T” refers to the total number of frames. However, audiences may likely think of them as different attention heads. Please consider a better way for the diagrams.*
> >
> > As a part of the rewriting of Section 2, we clarified what T and the plates from the diagram mean in the text. We hope that this whole section is now clearer.
> >
> > *Section 2.1 - Multi-head SummaryMixing. Please better explain the heads in SummaryMixing. In self-attention, heads are operating in parallel with different sets of parameters. However, here heads help to reduce the number of parameters. Please consider adding a diagram on this if it helps. In addition, by dividing x_t into n chunks, is it similar to chuck-wise attention or attentions with a limited context window? If that’s the case, please avoid using “head” here.*
> >
> > We agree with the reviewer that the use of the term “heads” here is misleading. Following the reviewer’s proposal, we also renamed it to “input chunking”, which is definitely more appropriate. We think that this is a relevant and improving engineering trick, but it is not mandatory to have it in the main article as long as it is properly mentioned. Therefore, we moved it to the Appendix. We also revised the text of this paragraph to make it easier to understand. We invite the reviewer to have a look at the new Appendix and let us know if any further clarifications are needed.
> >
> > *Section 3.3.1: The results shown in this section have a transformer decoder of 6-blocks-MHSA paired with each system. In this setup, the decoder will capture the global context, even if the encoder doesn’t. However, we can only conclude that self-attention is redundant when self-attention-based decoders are used. [...] However, there are two more problems to figure out. 1. How does vanilla Conformer/Transformer with CTC performance in this setup? They are the most common architectures for productions than Branchformer. 2. These results are obtained with LM shallow fusion, which can be expensive for deployment. How would the proposed approach perform without shallow fusion? If the language information is needed, please also consider RNN-T decoder.*
> >
> > We totally agree with the reviewer regarding their first question. This is why we ran a set of new experiments without any attentional decoder and with only a CTC decoder whose results are now exposed in section 3.3.1 and in the top comment. We wish also to highlight that we reported results for a CTC only decoding in the same section.
> >
> > From the results, we can observe that SummaryMixing achieves the same performance as MHSA with the Conformer architecture. This implies that, in this setup at least, self-attention is redundant and can be replaced with SummaryMixing, even without any attentional decoder. For the second question, we wish to mention that shallow fusion is not used at all in the results presented in Table 2, with 5 different ASR datasets. In the latter set of results, a simple beam search is applied, without any language model – and SummaryMixing consistently outperforms MHSA. Finally, given the findings observed both with attentional decoders as well as CTC only decoder, we expect an RNN-T system to follow the same path as the Transducer architecture does not contain any peculiarity that would make MHSA more relevant than in an encoder-decoder ASR system. From the theoretical perspective, nothing prevents us from using SummaryMixing with RNN-T. The only difference, however, would be that a SummaryMixing RNN-T should be able to handle streaming ASR as well, as it represents the most common use case of RNN-T ASR systems. While turning SummaryMixing into a streaming-capable cell is simple, and we are more than happy to discuss it here, building an entire experimental protocol for streaming ASR and RNN-T and adding it in a 9-page article appears to be irrealistic given the time frame of the discussion period. However, yes, SummaryMixing can be applied to RNN-T ASR.
> >
> > Thank you again to the reviewer for the remarks, comments, and questions. We would be happy to engage in a discussion and hope that our answers will have a positive impact on the final score.

---

> > > ### Author Response · Authors · 2023-11-21
> > > **End of the discussion period**
> > >
> > > Dear reviewer,
> > >
> > > With the deadline for the discussion period approaching very quickly (less than 48 hours remaining), we would like to know if our answers addressed your concerns or if further changes/answers should be provided.
> > >
> > > Many thanks again,
> > >
> > > The authors.

---

> > > ### Comment · Reviewer_gqu7 · 2023-11-21
> > > **Thanks for the clarifications!**
> > >
> > > I would like to appreciate the clarifications and improvements made to the manuscript. I updated my scores accordingly for the improvements.

---

> > > > ### Author Response · Authors · 2023-11-22
> > > > **Thank you, what else can we do?**
> > > >
> > > > Dear reviewer,
> > > >
> > > > Thank you for acknowledging the changes and increasing your score. We wish however to ask if anything else can be done that would make the reviewer think that SummaryMixing should be accepted. Indeed, a score of 5 remains a rejection. From our perspective, we answered all the reviewer's concerns and updated the paper accordingly, and we, therefore, do not know what could further be done to make this manuscript acceptable from the reviewer's perspective, whether it is for this conference or the next one. We still have a bit of time to update the manuscript and would be more than happy to try and fix any necessary changes leading to a potential acceptance.
> > > >
> > > > Many thanks again,
> > > >
> > > > The authors.

---

### Official Review · Reviewer_yNSy · 2023-10-29

**Soundness:** 3 good
**Presentation:** 2 fair
**Contribution:** 2 fair
**Rating:** 6
**Confidence:** 4

**Summary:**

This paper proposes a novel method to replace self-attentions with a more computationally efficient summary mixing model for speech recognition and spoken language understanding. Rather than computing an NxN attention matrix, the authors propose to first compute the temporal mean of the sequence and combine that with local information extracted using a cgMLP branch. Experiments on multiple corpora show that the proposed approach can match the performance of standard self-attention and Branchformer.

**Strengths:**

1. The paper uses a very simple approach to substitute self-attention with attention between a temporal summary and the original vector that appears to work well in practice.

2. Experiments are performed on multiple corpora for ASR (CommonVoice-Dutch, Italian, French), Librispeech, AISHELL, TED-LIUM 2, GSC and SLURP for SLU.

**Weaknesses:**

1. The paper makes the assumption that MHSA in the encoder is not necessary based on Zhang et. al. and Peng. et. al. However, I believe the assumption may not be well supported because (a) Zhang et al point out that you don't need self-attentions in higher layers, not that you don't need self-attentions at all, and (b) Peng. et al which uses self-attentions to model global context while using cgMLP to model local context. Therefore, the rationale behind this fundamental assumption made in the paper seems unclear.

2. SLURP uses read speech - the impact of temporal averaging here may be minimal, but may be very harmful in spontaneous conversational speech. Such explorations in my view are important in this work to claim that SummaryMixing can reach top-of-the-line in speech understanding.

3. Writing could be more self-contained and clear in certain places - for example, HyperMixer is not well described which makes it hard to assess the relationship between the methods.

**Questions:**

1. The authors claim that a score of 0.5 diagonality implies a uniform distribution and therefore, a simple average could achieve a similar token mixing. However, this is not obvious or clear to me. Could the authors explain?

2. I was interested to know how the performance of summary mixing is impacted during inference by temporal averaging. As the sequence length increases, the sum vector becomes a mixture of more frames, degrading the ability to discriminate individual frames/tokens. Do the authors have any numbers showing ASR performance (WER) as a function of sequence length across models with self-attention and the proposed SummaryMixing ?

3. As SLURP uses read speech, I was wondering if the authors had performed experiments on other corpora with spontaneous speech, for example, SLUE-VoxPopuli [1] for Named Entity Recognition (NER)?

4. The paper mentions KWS results, but I don't see any in the results section.

5. There are some papers [2,3,4] that show linear transformers can obtain comparable performance to MHSA for ASR contrary to what the authors claim. These works are also relevant and must be acknowledged in the paper.

[1] "SLUE Phase-2: A Benchmark Suite of Diverse Spoken Language Understanding Tasks",
Suwon Shon, Siddhant Arora, Chyi-Jiunn Lin, Ankita Pasad, Felix Wu, Roshan S Sharma, Wei-Lun Wu, Hung-yi Lee, Karen Livescu, Shinji Watanabe

[2] Jingyu Sun, Guiping Zhong, Dinghao Zhou, Baoxiang Li, Yiran Zhong, "Locality Matters: A Locality-Biased Linear Attention for Automatic Speech Recognition"

[3] "Conformer-Based Speech Recognition with Linear Nyström Attention and Rotary Position Embedding",
Lahiru Samarakoon, Tsun-Yat Leung

[4] "Fast Conformer with Linearly Scalable Attention for Efficient Speech Recognition", Dima Rekesh et. al.

---

> ### Author Response · Authors · 2023-11-16
> **Thank you 1/2.**
>
> We would like to thank the reviewer for taking the time to read and review this article. We are glad to read that the reviewer found SummaryMixing to be a simple yet effective approach to replace MHSA. We wish to mention that we can provide an anonymized version to the reviewer if necessary. The concerns and remarks of the reviewer are addressed in the following:
>
> **Weaknesses**
>
> *The paper makes the assumption that MHSA in the encoder is not necessary based on Zhang et. al. and Peng. et. al. [...] Zhang et al point out that you don't need self-attentions in higher layers, not that you don't need self-attentions at all [...] Peng. et al which uses self-attentions to model global context while using cgMLP to model local context. Therefore, the rationale behind this fundamental assumption made in the paper seems unclear.*
>
> We agree with the reviewer that this motivation is key. This is why we changed section 1 to better detail this motivation. To be more precise and answer this question, we would like to highlight that with this article, our motivation is that both works seem to indicate that self-attention **may** not be useful under certain conditions that the reviewer mentioned. Before experimenting, we did not know that self-attention could be entirely replaced by SummaryMixing – the literature just gave us a hint in that direction, and that is why the paper decided to propose and explore SummaryMixing. In the Branchformer paper, Figure 6 clearly indicates that the self-attention weights are close to being uniformly distributed from the very first layer to the last layer. Based on the definition of “diagonality” in Figure 6 of the Branchformer paper, diagonality 0.5 implies weights are uniformly distributed, and almost all layers have diagonality between 0.5 and 0.6. Also, uniformly distributed attention weights imply that the weighted sum w.r.t. weights is equal to (up to a constant factor) an unweighted sum. Although the attention weights are not strictly uniformly distributed based on Figure 6, it gives us hints to propose the SummaryMixing operation. Also, although the diagonality is measured for Branchformer, we also test SumamryMixing for Conformer in our paper.
>
> *SLURP uses read speech - the impact of temporal averaging here may be minimal, but may be very harmful in spontaneous conversational speech. Such explorations in my view are important in this work to claim that SummaryMixing can reach top-of-the-line in speech understanding.*
>
> We would like to address this point in two parts. First, and following the reviewer’s request, we conducted an experiment on a very noisy dataset with conversational speech: AMI. We wish to mention that there is no AMI ASR recipe available in SpeechBrain yet, and we tried our best based on the literature and the results are not far away from the numbers in other tools, like K2 or ESPnet.  From the results, the reviewer can see that SummaryMixing still works competitively compared to MHSA. It also performs much better than the previous best baseline: the Fastformer linear operation baseline fails to reach the same level of performance.
>
>  **WERs on AMI dataset (conversational and noisy speech)**
> | Model (Conformer Encoder-Decoder CTC+Attention) | Dev | Test |
> | -------- | -------- | -------- |
> | MHSA (41.4M)    | 21.9 | **18.2**|
> | SummaryMixing (39.9M) | **21.8**| 19.4|
> | Fastformer (39.9M) | 25.1 | 21.3|
>
> Second, ASR is different from SLU, and we therefore can not affirm that such findings could be extended to SLU. As a matter of fact, we decided to follow the same experimental protocol as the [Branchformer paper](https://arxiv.org/pdf/2207.02971.pdf) accepted at ICML, hence the choice of the SLURP and Google Speech Command datasets. AMI was the only available (commercially exploitable and in SpeechBrain) dataset that we could use to try quickly SummaryMixing on very noisy and conversational speech.

---

> > ### Author Response · Authors · 2023-11-16
> > **Thank you 2/2.**
> >
> > **Questions**
> >
> > *The authors claim that a score of 0.5 diagonality implies a uniform distribution and therefore, a simple average could achieve a similar token mixing. However, this is not obvious or clear to me. Could the authors explain?*
> >
> > Given this comment, we revised section 1 to add more details about this issue. We would like to answer this more in detail here as well. Based on the definition of diagonality of Zhang. et. al. and Peng. et. al. , diagonality 0 indicates all the weights are pushed as far away as possible from the diagonal, diagonality 0.5 means all the weights are uniformly distributed along the sequence and diagonality 1 implies all the weights are distributed along the diagonal. For the detailed formula of “diagonality”, we invite the author to look at Zhang. et. al. (arxiv.org/pdf/2011.04906.pdf) or Peng. et. al. (https://arxiv.org/pdf/2207.02971.pdf)
> >
> >
> > *I was interested to know how the performance of summary mixing is impacted during inference by temporal averaging. As the sequence length increases, the sum vector becomes a mixture of more frames, degrading the ability to discriminate individual frames/tokens. Do the authors have any numbers showing ASR performance (WER) as a function of sequence length across models with self-attention and the proposed SummaryMixing ?*
> >
> > Following the advice of the reviewer, we added new results (in the Appendix and in the top message of these reviews) showing the decoding WER of SummaryMixing, Fastformer and MHSA as a function of the sequence length. It appears that SummaryMixing is not more negatively impacted by the sequence length than MHSA. It is actually the opposite as MHSA has a higher increase in WER following the sentence duration.
> >
> > *As SLURP uses read speech, I was wondering if the authors had performed experiments on other corpora with spontaneous speech, for example, SLUE-VoxPopuli [1] for Named Entity Recognition (NER)?*
> >
> > We again agree with the reviewer. In a previous answer (see weaknesses section), we mentioned the new results obtained on the noisy and conversational speech AMI dataset. SummaryMixing still works just as well as self-attention, even in this challenging environment. We also had a look at the available recipes in SpeechBrain that we could run on more complex SLU datasets, unfortunately, we could not find any.
> >
> > *The paper mentions KWS results, but I don't see any in the results section.*
> >
> > Table 2 reports the accuracies of the different models on the Google Speech Command dataset. We agree with the reviewer that this dataset qualifies a bit loosely to the definition of Keyword Spotting, however, this is how Google themselves [qualify this dataset](https://www.tensorflow.org/datasets/catalog/speech_commands?hl=en).
> >
> > *There are some papers [2,3,4] that show linear transformers can obtain comparable performance to MHSA for ASR contrary to what the authors claim. These works are also relevant and must be acknowledged in the paper.*
> >
> > First of all, we agree that other linear alternatives to MHSA exist in the literature. We also understand that this might not be clear enough from the paper and this is why we modified the paper accordingly. We invite the reviewer to have a look at Section 1.1 to make sure that this now appears clearly. [2, 3] indeed passed below our radars, and we must acknowledge it (as we did in Section 1.1). We wish to mention however that no code is given to reproduce the result, while SummaryMixing is already out and usable in SpeechBrain. Also, in [3], the authors wish to keep the overall pair-wise interactions of self-attention, they simply restrict the window with a clever solution. SummaryMixing is fundamentally different – it completely removes self-attention. Finally [4] does not have linear attention at training time, the authors propose to apply a window to self-attention post-training to limit the decoding time increase with respect to the sequence length, they also use aggressive downsampling to reduce the inference time. SummaryMixing is linear at training time and decoding time, without any trick. Finally, we wish to highlight the fact that one of our baseline, Fastformer, was in fact compared to many linear SOTA alternatives to self-attention in speech. Fastformer always beats the other methods. This is why we decided to pick Fastformer as a baseline, because beating it would imply the superiority of SummaryMixing over the other techniques as well, at least in the context of the environment protocols defined in the original Fastformer paper in ours.
> >
> > We wish to thank again the reviewer for taking the time to review this work and provide both positive and negative feedback. We hope to engage with the reviewer during the discussion period and that our answers will have a positive impact on the final score.

---

> > > ### Author Response · Authors · 2023-11-21
> > > **End of the discussion period**
> > >
> > > Dear reviewer,
> > >
> > > With the deadline for the discussion period approaching very quickly (less than 48 hours remaining), we would like to know if our answers addressed your concerns or if further changes/answers should be provided.
> > >
> > > Many thanks again,
> > >
> > > The authors.

---

> > > > ### Comment · Reviewer_yNSy · 2023-11-21
> > > >
> > > > I thank the reviewers for their responses.
> > > >
> > > >
> > > > Re. motivation, the following is not clear to me - could you explain ?
> > > >
> > > > ```
> > > > Also, uniformly distributed attention weights imply that the weighted sum w.r.t. weights is equal to (up to a constant factor) an unweighted sum. Although the attention weights are not strictly uniformly distributed based on Figure 6, it gives us hints to propose the SummaryMixing operation.
> > > > ```
> > > > I understand the progression of events better now, and it seems like this motivation only supports a modification of the Branchformer.
> > > >
> > > > I appreciate the additional AMI results, but it would have been ideal to do such experiments on speech understanding corpora. I encourage authors to try and do this experiment and add it to the camera ready if accepted.
> > > >
> > > > I appreciate the additional analysis on temporal averaging and the note on KWS. The authors have also added some information on related work in ASR.
> > > >
> > > > About linear transformers: The cosformer used by Sun. et. al does, in fact, appears to match or beat the standard conformer results from their paper, and in my mind, is a state-of-the-art baseline for speech[1] and non-speech[2] tasks. The code for the self-attention implementation is also public here[3]. This would have been good to compare to.
> > > >
> > > > Based on the provided responses, I have updated my score.
> > > > I would like to encourage the authors if given a chance to conduct more expansive tests across speech tasks like speech summarization, and non-speech tasks like Wiki-LM to make this paper more relevant and useful to broader audiences.
> > > >
> > > >
> > > > [1] Jingyu Sun, Guiping Zhong, Dinghao Zhou, Baoxiang Li, Yiran Zhong, "Locality Matters: A Locality-Biased Linear Attention for Automatic Speech Recognition"
> > > > [2] "cosFormer: Rethinking Softmax in Attention", Zhen Qin, Weixuan Sun, Hui Deng, Dongxu Li, Yunshen Wei, Baohong Lv, Junjie Yan, Lingpeng Kong, Yiran Zhong, ICLR 2022
> > > > [3] https://github.com/OpenNLPLab/cosFormer

---

> ### Author Response · Authors · 2023-11-22
> **Answering the last few questions**
>
> Dear reviewer,
>
> We wish to thank you for acknowledging the changes and updating your score.
>
> First, we would like to clarify the motivation issue. Branchformer is not the only architecture showing MHSA layers behave like linear operations. In the third paragraph of Section 1, we mention works that show some layers in Transformers and Conformer also behave as linear operations. This is what motivated us to try SummaryMixing with both Branchformer and Conformer which are, currently, the two state-of-the-art and most common speech encoders. We hope that this clarifies the motivation. If not, please let us know and we will extend the answer further.
>
> Second, we wish to address the request for more experiments across tasks. Another work submitted to ICLR 2024 trying to speed up ASR training (and ASR only,) under a much more restricted experimental scenario (only 3 datasets, and we use 2 of them) than SummaryMixing, has received excellent scores without any mention of a potential scope issue or audience issue. Both models are not strictly comparable -- their is much more complex to implement and use and still has a quadratic-time complexity while achieving similar WER performance to ours. However, we must remark that the datasets and tasks that we already cover in SummaryMixing are more numerous than this other submission which will most likely be accepted according to the reviewers.  With that being said, we are now targeting a few more experiments as mentioned by the reviewer, but we of course can't give any results before the deadline of the discussion period.
>
> Finally, we agree with the reviewer on the cosFormer -- and we will try it for the camera-ready version (if accepted). We also wish to highlight that the new Related Work now includes a mention to "Locality Matters: A Locality-Biased Linear Attention for Automatic Speech Recognition".
>
> We thank again the reviewer for the discussion, and we remain available until the end of the rebuttal period (24 hours away).
>
> The authors.

---

### Official Review · Reviewer_fkrd · 2023-11-01

**Soundness:** 4 excellent
**Presentation:** 3 good
**Contribution:** 3 good
**Rating:** 8
**Confidence:** 5

**Summary:**

The paper proposed a global summarization layer as a replacement for MHSA in several ASR architectures such as branchformer and Conformer. This layer or module is introduced and compared with several parallel implementation in an open source project SpeechBrain.
The newly proposed component dubbed summaryMixing has a linear computation cost on the encoder ASR components.
Expectation is that this will close the gap with MHSA but actually authors found this surpasses it in their tests.

**Strengths:**

The paper proposes a generalization of hyperMixer and apply it to Speech tasks.
The generalization is simple but its effectiveness is surprising.
The proposed architecture works very well in comparison with many baselines and architectures (Conformer, Branchformer, ContextNet, Branchformer +fastattention, Branchformer w/o attention, ...)
The obtained and reported results are very competitive with various ASR benchmarks and setups.
It is worth noting the surprisingly good performance of one of the simplest baseline, the branchformer w/o attention, which authors acknowledge and highlight.
They consider encoder architecture only to show the unnecessary MHSA complexity for ASR tasks is not coming exclusively from the top decoder..

**Weaknesses:**

There is a couple of points where the newly proposed component experimentation could have been improved.
The authors should clarify the ASR architecture in the final version since the current description of the ASR architecture is vague and prone to confusions-- see  questions below --.  I think this is something that authors might be aware of but prefer refer readers to the SpeechBrain recipes and experimentation.  This might leave a shallow reader without immediate interest in reproducing or delving into the code and recipes with a confusing architecture.

The approach has some limitations, so it would be intriguing to examine the performance of the proposed encoder in long-context ASR.
Since this is a leaning representation conference, some experiments on NLP tasks, as in the original HyperMixer paper, would have attracted more readers.

Given the nature of speed optimization and comparison with MHSA, it would be interesting for the readers to know the specifics of the MHSA implementation as there are many very efficient implementations available nowadays.

Finally, it would have been nice to compare the proposed approach in both batch and online ASR.

**Questions:**

The following questions might need some attention into the manuscript:
* which is the global architecture and loss ? It is clear that it is a encoder decoder based architecture with CTC loss, but this is an important detail mentioned in 1 line. How many encoder layers each arch has ? which are the details of the decoder ?
* Could the MHSA improve as the context becomes larger ?
* which is the MSHA implementation have you used ?

---

> ### Author Response · Authors · 2023-11-16
> **Thank you 1/2.**
>
> We wish to thank the reviewer for highlighting the effectiveness of the proposed method and mentioning that the results are very competitive with various ASR benchmarks and setups. We are glad to see that the reviewer finds the results interesting and appealing to the community. We wish to mention that we can provide an anonymized version to the reviewer if necessary. In the following, we address the remaining comments and questions of the reviewers.
>
> **Weaknesses**
>
> *I think this is something that authors might be aware of but prefer refer readers to the SpeechBrain recipes and experimentation. This might leave a shallow reader without immediate interest in reproducing or delving into the code and recipes with a confusing architecture.*
>
> We totally agree with the reviewer. The reason for this design choice is that, for instance, the yaml (hyperparameters definition) of the model trained on Librispeech that we have released on GitHub (please do not look for it, it’s not anonymized, we can provide it here if necessary) contains 260 lines of model definition. Hence, we initially thought that this would be better for the reader to just have access to our parameter files (**available on Github for all experiments**) as well as the SpeechBrain recipes would be sufficient. However, we added a simplified version of the model definition that hopefully will help the reader to get a better understanding of the model architecture in the Appendix. We invite the reviewer to have a read at the Appendix and let us know if any more information should be added.
>
> *The approach has some limitations, so it would be intriguing to examine the performance of the proposed encoder in long-context ASR. Since this is a learning representation conference, some experiments on NLP tasks, as in the original HyperMixer paper, would have attracted more readers.*
>
> We agree with the reviewer and invite them to have a look at the top comment illustrating the new results on the evolution of the WER as a function of the sequence length of SummaryMixing vs Fastformer vs MHSA. We also added these results to the Appendix of the manuscript. From the results, it is clear that SummaryMixing does not suffer more than MHSA from longer sentences, it actually looks like the opposite. We also share the reviewer’s view on the application domain. However, our primary domain of expertise is speech processing, and it is unclear if 10 days would be sufficient for us to build a trustworthy experimental protocol on NLP tasks.
>
> *Given the nature of speed optimization and comparison with MHSA, it would be interesting for the readers to know the specifics of the MHSA implementation as there are many very efficient implementations available nowadays.*
>
> We agree that this is not clear from the current form of the manuscript. We invite the reviewer to have a look at section 3.1 to make sure that the revised article solves this issue. In practice, the used MHSA implementation is the one from SpeechBrain (with relative positional encoding for better ASR performance) implemented in PyTorch [here](https://github.com/speechbrain/speechbrain/blob/bd27e99119edfe6b4f530be16d46cce28af260b3/speechbrain/nnet/attention.py#L362). For the speed perspective, we compared it against the regular MHSA from PyTorch (visible [here](https://github.com/speechbrain/speechbrain/blob/bd27e99119edfe6b4f530be16d46cce28af260b3/speechbrain/nnet/attention.py#L642)  in SpeechBrain) and found no significant degradation in speed. We however did not compare this to other forms of very efficient MHSA implementation such as FlashAttention as they benefit from optimisations linked to the compilation and it would be unfair to compare this with SummaryMixing that is a raw PyTorch implementation.
>
> *Finally, it would have been nice to compare the proposed approach in both batch and online ASR.*
>
> For now, we agree that only offline ASR is addressed. The primary reason for this choice is that, at the time of writing, the only available online ASR pipeline in SpeechBrain is still under Pull Request. We also fully understand that this is not a fully valid answer for not trying it on K2 for instance, but the latter would also require a complete change of environment, with different baselines (e.g. fastformer, hypermixer, the branchformer, and context net does not exist in K2). This is not realistic for a 9 page articles and within 10 days.

---

> > ### Author Response · Authors · 2023-11-16
> > **Thank you 2/2**
> >
> > **Questions**
> >
> > *Which is the global architecture and loss ? It is clear that it is a encoder decoder based architecture with CTC loss, but this is an important detail mentioned in 1 line. How many encoder layers each arch has? which are the details of the decoder ?*
> >
> > We agree with the reviewer that the lack of such information is problematic. We clarified these points in the main paper in section 3.1 and in Appendix A.6. The global architecture is encoder-decoder with CTC plus Attention training. The encoder (conformer or branchformer) is attached to a CTC decoder and a Transformer decoder. The loss is the standard CTC+Attention loss and is typically visible [here](https://github.com/speechbrain/speechbrain/blob/bd27e99119edfe6b4f530be16d46cce28af260b3/recipes/LibriSpeech/ASR/transformer/train.py#L122, this code is not from us, so not breaking any anonymity).  The number of layers depends on the dataset and the size of the model, but it ranges from 12 to 18. The decoders are always the same – a simple dense layer for CTC and 4 or 6 transformer decoder layers depending on the model size. These parameters are entirely described in the SpeechBrain configuration files that we released (and that we can share anonymously here as well), such as **num_decoder_layers: 6**.
> >
> > *Could the MHSA improve as the context becomes larger ?*
> >
> > We also think that this question is interesting. Therefore, we added an experiment to the Appendix as well as the top message of this thread showing the evolution of the WER following the increase in audio duration for MHSA, SummaryMixing and Fastformer. SummaryMixing does not suffer more than MHSA from increased sentence length. Therefore, It is most likely that MHSA will not improve much over SummaryMixing with longer sentences – at least in typical speech-related tasks.
> >
> > *Which is the MSHA implementation have you used?*
> >
> > We detailed this answer in the “Weaknesses” sub-section of this rebuttal. In short: the SpeechBrain i.e. PyTorch implementation of relative positional encoding self-attention.
> >
> > We wish to thank again the reviewer for the positive and negative feedback. We hope to engage with the reviewer during the discussion period and that our answers will have a positive impact on the final score.

---

> > > ### Author Response · Authors · 2023-11-21
> > > **End of the discussion period**
> > >
> > > Dear reviewer,
> > >
> > > With the deadline for the discussion period approaching very quickly (less than 48 hours remaining), we would like to know if our answers addressed your concerns or if further changes/answers should be provided.
> > >
> > > Many thanks again,
> > >
> > > The authors.

---

### Official Review · Reviewer_wGqz · 2023-11-05

**Soundness:** 3 good
**Presentation:** 2 fair
**Contribution:** 3 good
**Rating:** 3
**Confidence:** 5

**Summary:**

This paper proposes a novel neural network architecture called SummaryMixing for speech recognition. The motivation of this work is to replace the heavy computational cost yielded by transformer self-attention blocks with a SummaryMixing block on top of the branchformer architecture. SummaryMixing shows effectiveness by offering a competitive performance from the original branchformer architecture, especially with the low computational cost, mainly within the CTC framework.

**Strengths:**

- Simple and yet another neural network architecture by extending the branchformer architecture to capture global characteristics via SummaryMixing and local characteristics via cgMLP.
- Showing the low computational cost compared with the original branchformer
- Good reproducibility (open source implementation and the use of public data).

**Weaknesses:**

- The survey of the efficient transformer (conformed) is not sufficient. There are a lot of efficient transformers in the NLP and ML field (e.g., https://arxiv.org/pdf/2009.06732.pdf). Even if we limit the discussions to speech applications, there are a lot of methods (e.g., squeezeformer, efficient conformer, emformer, etc.).
- The performance improvement from the conventional method (e.g., fastformer) is marginal.
- The method is on top of branchformer, which already uses Fastformer, and its technical novelty is weak.
- Several descriptions (e.g., the surveys, distinction from HyperMixer/MLP Mixer, and detailed architectures) are unclear.
  - Frankly, I could not understand the distinction of the SummaryMixing block from HyperMixer/MLP Mixer only from Section 2.1, mainly due to the lack of an explanation of what HyperMixer/MLP Mixer is.
  - It is difficult to understand the detailed architectures only from this description: "In particular, the transformation (f), summary (s), and combiner (c) functions are all implemented as a dense linear layer followed by a GeLU activation function." Similarly, how the model size of the proposed and other methods is adjusted was unclear.

**Questions:**

- Can you expand the discussion of how this novel architecture and findings would attract the general AI and ML researchers in ICLR? This paper specializes in speech recognition (and spoken language understanding, which is a very similar task to speech recognition), and it is a narrow scope for me.
- Besides the above expansion, can you explain why you selected Fastformermer and ContextNet?
- Can you apply this to RNN-T?

Other minor comments
- Abstract: ASR --> automatic speech recognition (ASR), speech understanding --> spoken language understanding
- Page 3, last paragraph "Hence, $X \in \mathbb{R}$ becomes $X \in \mathbb{R} ^{T \times D/n}$": Is $X \in \mathbb{R}$ scalar? I think you missed adding some domains.

---

> ### Author Response · Authors · 2023-11-16
> **Thank you 1/2.**
>
> We first would like to thank the reviewer for highlighting the good reproducibility of SummaryMixing as well as its low computational cost. We wish to mention that we can provide an anonymized version to the reviewer if necessary. In the following, we address every single comment made by the reviewer and answer the given questions.
>
> **Weaknesses**
>
> *The survey of the efficient transformer (conformed) is not sufficient. [...] Even if we limit the discussions to speech applications, there are a lot of methods*
>
> We agree with the reviewer that comparing with more “linear attention” is important and we modified the related work (Section 1.1) of the manuscript accordingly to incorporate many relevant methods. We invite the reviewer to take a look at this revised section. Experiment-wise, however, It is worth mentioning that our SummaryMixing was compared with two “linear-attention” alternatives - Fastformer and Hypermixer. In the original paper, Fastformer has been compared to longformer, Linformer, linear transformers, BigBird and poolingformer. All these methods are state-of-the-art alternatives to self-attention. In this original published work, Fastformer was better than all of them and has been replicated in ESPNet. This is why it appeared as a strong baseline in our case. Indeed, if Fastformer beats most existing linear alternatives, our goal should be to beat Fastformer.  Then, we wish to highlight that there exists no available implementation of any linear-attention claiming SOTA performance on ASR benchmarks -- SummaryMixing is the first one. Fastformer is implemented in ESPnet, but does not reach SOTA – SummaryMixing does on most if not all ASR datasets.
>
> *The performance improvement from the conventional method (e.g., fastformer) is marginal.*
>
> The Fastformer also can not catch the performance of MHSA, hence compute and memory costs were coming at the cost of accuracy. This is not true with SummaryMixing as it reaches or even beats MHSA. We agree with the reviewer that absolute improvements do not seem vastly game-changing. However, we wish to highlight that the BranchFormer, which simply makes the conformer parallel instead of sequential and was published at ICML, improved the performance over the Conformer by .1% on Librispeech dev-clean and .2% on test-clean. Compared to Fastformer; SummaryMixing improves the performance by .2% .1% .3% on Librispeech dev-clean, test-clean and test-others respectively. if we consider other ASR tasks, SummaryMixing always beats Fastformer, with an average absolute gain of 0.75%. The latter improvement is more significant than the one originating from Branchformer vs Conformer from the reference ICML paper. The performance might appear marginal in the reviewer’s perspective, which might be true, yet it is consistent, and more importantly, totally aligned with what we see from the improvements in ASR from the literature.
>
> *The method is on top of branchformer, which already uses Fastformer, and its technical novelty is weak.*
>
> We modified Section 2.2 to clarify the use of SummaryMixing both in Conformer and Branchformer. Also, we wish to bring our perspective to the potential lack of novelty. If we see things from the Branchformer point of view, we agree with the reviewer, in proportion, the Sumformer implies only a few changes (and this is a strength from our point of view). However, we would like to first highlight that SummaryMixing is also applied to the Conformer in this paper (Table 1 and CTC only results) and it also beats or matches MHSA. SummaryMixing is expected to replace MHSA in any speech encoder equipped with it, and this article demonstrates that it works just better with the two most used architectures: Conformer and Branchformer. Second, we wish to mention that Branchformer was accepted at ICML while its contribution was to make two blocks of layers parallel instead of sequential – from our very own point of view, this does not represent much more novelty than SummaryMixing (quantitatively speaking), yet it was accepted and is now used by the community.
>
> *Several descriptions (e.g., the surveys, distinction from HyperMixer/MLP Mixer, and detailed architectures) are unclear.*
>
> We thank the reviewer for mentioning these issues. We revised the paper according to the three problems mentioned by the reviewer, see Section 1.1 for the survey, Section 2 and the Appendix for the Hypermixer, and Section 3.1 and the Appendix for the detailed architecture. Please let us know if the reviewer still finds the definitions confusing or lacking precision, as this only concerns text, we can easily fix it and align it with the reviewer's perspective.

---

> > ### Author Response · Authors · 2023-11-16
> > **Thank you 2/2**
> >
> > **Questions**
> >
> > *Can you expand the discussion of how this novel architecture and findings would attract the general AI and ML researchers in ICLR? This paper specializes in speech recognition (and spoken language understanding, which is a very similar task to speech recognition), and it is a narrow scope for me.*
> >
> > We wish to answer the question in two parts. First, we wish to highlight that the ICLR call for papers clearly mentions that works focusing on applications to audio/language are welcome (“Representation learning for computer vision, audio, language, and other modalities”). Furthermore, plenty of articles have been published in conferences like NeurIPS / ICLR or ICML while only focusing on speech or specific application domains. For instance “Squeezeformer: An Efficient Transformer for Automatic Speech Recognition” was accepted at NeurIPS 2022 and only used CTC models on a single dataset (Librispeech). This is also true for “Understanding The Role Of Self Attention For Efficient Speech Recognition” accepted at ICLR 2022, only using the Conformer-CTC only model and the Librispeech dataset. Another example is the Branchformer, which only was evaluated on speech recognition and SLU, just like SummaryMixing, was published at ICML 2022. On other domains, MLP Mixer was strictly designed for and evaluated on clean and simple vision tasks yet published at NeurIPS 2021. MLP Mixer was, by design, not usable for variable length sequences including most speech and NLP tasks. Second, we still wish to answer the question properly. We believe that SummaryMixing would attract interest, or at least raise discussions, within the general ML community as SummaryMixing is a replacement to MHSA, evaluated on speech, but that could be extended to other modalities, very much like MLP mixer which was only designed and evaluated for vision. SummaryMixing was conceived with speech in mind because the latter domains fall within our expertise and tools, but it could be applied to vision or NLP by researchers who possess the level of knowledge in such domains.
> >
> >
> > *Besides the above expansion, can you explain why you selected Fastformermer and ContextNet?*
> >
> > We agree that the reasons for this choice are not clear. Hence, we modified Section 1.1 as well as Section 3.1 to better motivate this decision. **In short, we selected both models as baselines as they, from the literature, represent the SOTA in the tasks that we selected to evaluate SummaryMixing.** More precisely, we selected Fastformer because the original paper compares it to many SOTA linear MHSA alternatives across different tasks. Fastformer always beat them both accuracy and speed-wise. Hence, if SummaryMixing is able to beat Fastformer (and it is), then we can simply deduce that it would beat the others as well, at the very least in the context of the typical ASR / SLU environmental protocols defined in our article and in the original Fastformer/Branchformer articles. ContextNet was added as it was (at the date of writing) the best CNN-only model for ASR. In practice, that is only true on paper as no one ever succeeded in reproducing the claimed results from the original paper, but the final performance that we obtained was already pretty close to SOTA with this architecture. ContextNet was also available on SpeechBrain, making it a good choice from the reproducibility perspective of our potential readers.
> >
> > *Can you apply this to RNN-T?*
> >
> > From the theoretical perspective, nothing prevents us from using SummaryMixing with RNN-T. The observed results from the article with CTC/Attention and CTC-only training clearly indicate that SummaryMixing works really well with these different decoders and the Transducer would not add any drastic change causing SummaryMixing to fail. The only difference, however, would be that a SummaryMixing RNN-T should be able to handle streaming ASR as well, as it represents the most common use case of RNN-T ASR systems. While turning SummaryMixing into a streaming-capable cell is simple, and we are more than happy to discuss it here. Building an entire experimental protocol for streaming ASR and RNN-T and adding it in a 9-page article appears to be unrealistic given the time frame of the discussion period. However, yes, SummaryMixing can be applied to RNN-T ASR.
> >
> >
> >
> > We wish to thank the reviewer again for the positive comments and remarks/questions/critique. We would be happy to engage in a discussion and hope that our answers will have a positive impact on the final score.

---

> > > ### Author Response · Authors · 2023-11-21
> > > **End of the discussion period.**
> > >
> > > Dear reviewer,
> > >
> > > With the deadline for the discussion period approaching very quickly (less than 48 hours remaining), we would like to know if our answers addressed your concerns or if further changes/answers should be provided.
> > >
> > > Many thanks again,
> > >
> > > The authors.

---

> > > > ### Author Response · Authors · 2023-11-22
> > > > **Additional comment on paper scope and ICLR.**
> > > >
> > > > Dear reviewer,
> > > >
> > > > As the deadline for the discussion is 24 hours away, we wanted to add one last comment about the reviewer's argument mentioning that *"This paper specializes in speech recognition (and spoken language understanding, which is a very similar task to speech recognition), and it is a narrow scope for me."*
> > > >
> > > > We already mentioned in the original answer that the call for papers to ICLR explicitly mentions Audio research, but also that another [submitted paper to ICLR 2024](https://openreview.net/forum?id=9WD9KwssyT) trying to speed up ASR training (and ASR only) under a much more restricted experimental scenario than SummaryMixing, has received excellent scores without any mention to a potential scope issue.
> > > >
> > > > Many thanks,
> > > >
> > > > The authors

---

### Comment · Area_Chair_Woa4 · 2023-11-10
**reviewer-author discussions**

Dear All,

The reviewer-author discussion period will be from Nov. 10 to Nov. 22. For reviewers, please read the authors' responses and acknowledge it, respond to them early on in the discussion, and discuss points of disagreement. Thank you!

AC

---

### Author Response · Authors · 2023-11-16
**General answer**

**Introduction**

First of all, we wish to sincerely thank the reviewers and AC for taking the time to assess this submission. We are also glad to read that reviewers find the results of SummaryMixing convincing and appealing to the community. We have now uploaded a new version of the manuscript (and the supplementary material) containing all the changes necessary to address the reviewer’s remarks and concerns. These changes are highlighted with a Lavender color. We also answered the reviewers individually to address every single comment/remark/question. This top message has two aims: (a) it summarises the major changes in the manuscript; (b) it presents two new experiments based on CTC-only training as well as an analysis of the word error rate as a function of the audio length.

**Major changes**

All the reported changes to the manuscript come from requests/remarks/questions of the reviewers.
- The relation between SummaryMixing and Hypermixing as well as the explanation of SummaryMixing have been entirely rewritten. This is visible in Section 2 and a new dedicated section in Appendix A.1 with new figures and an exhaustive definition.
- Section 3.3.1 now contains a **new experiment** on CTC-only training (no attention at all).
- Appendix A.5 now contains a **new experiment** analyzing the impact of the audio length on the word error rate.
- A new paragraph has been added to the introduction to better motivate the use of an average (hence SummaryMixing). This paragraph also makes more explicit what a diagonality score of 0.5 means.
Half a page of ASR and linear attention for ASR related works have been added to section 1.2. This change includes the various references proposed by the reviewers.
- The experimental protocol now better details the ASR architecture (type of ASR, implementation details) as well as the implementation of MHSA. We also added 5 tables describing **all the hyperparameters** necessary to reproduce all the ASR experiments to Appendix A.6.
- The “attention-head for SummaryMixing” paragraph has been renamed to “input chunking” and moved to the Appendix.
Section 2.2 now clearly explains that SummaryMixing is applied to both the Branchformer and the Conformer architecture. Appendix A.4 also details this change with a figure.

**CTC-only training experiment**

We performed CTC-only training on Librispeech with a Conformer encoder ASR equipped with MHSA and SummaryMixing. In this scenario, the decoder is not able to recover from the lack of self-attention in the encoder part as it only is a dense layer and CTC. The full set of hyperparameters is given in Appendix A.6.

 *CTC greedy decoding WERs on the LibriSpeech dataset*
| Model (Conformer CTC) | dev-clean | test-clean | test-other |
| -------- | -------- | -------- | -------- |
| MHSA (28.8M)    |  3.5  | 3.7 | 9.2|
| SummaryMixing (26.5M) | 3.5| 3.7 | 9.4|

From the results, it is quite clear that SummaryMixing can perform just as well as MHSA, even without any self-attention in the whole architecture – *at least on the Librispeech dataset*.

**Experiment: WER as a function of audio length**

The sensitivity of SummaryMixing, Fastformer, and self-attention to the variation of the duration of audio files during speech recognition decoding is compared in this experiment. To achieve this, we evaluate the WER of the small Branchformers trained on the Tedlium 2 dataset and presented in Table 2 on ten sets of sentences of increasing duration. As a reminder, this ASR model is a Branchformer encoder with a transformer decoder trained jointly with CTC and without any language model. These sets are designed by taking the test set of Tedlium and splitting it into 10 partitions where sentences fall into buckets of corresponding lengths. We then compute the WER of the Branchformers equipped with SummaryMixing, SummaryMixing-lite, Fastformer, and self-attention and report the result for each bucket of increasing duration in the given Figure or in **Figure 7 in Appendix A.5**.

[Click here to see the results of this new experiment or go to the Figure 7 in Appendix A.5](https://i.ibb.co/QDmtvzp/wer-length.png)

From the results, it is clear that not only both SummaryMixing and SummaryMixing-lite perform the best, but also that longer sentences do not harm SummaryMixing more than MHSA. It appears to be the opposite as the WER increases more rapidly for MHSA than SummaryMixing with the increase in audio duration. Hence, we can conclude that SummaryMixing does not alter the long-term context learning capabilities of encoder-decoder ASR systems when replacing MHSA.

**Conclusion**

Again, we wish to thank everyone involved in the review process. With this rebuttal and updated submission, we answered every single comment/remark/question of the reviewers. We hope that this will either lead to further interesting discussions or to an increase in the scores.

---

### Meta-Review · Area_Chair_Woa4 · 2023-12-04

**Metareview:**

The paper proposes a novel method called SummaryMixing, which replaces self-attention with a mean-based summary of the whole utterance. This method reduces the computational and memory costs of speech processing models, while preserving or exceeding the accuracy of self-attention models. The proposed SummaryMixing method has been applied to Conformer and Branchformer to show its effectiveness. The paper also reports experimental results on several speech processing tasks, such as ASR and SLU, and compares SummaryMixing with self-attention and other alternatives. The authors have been very active in participating in the author-reviewers discussion, with the diligence to address the concerns from all reviewers.

Sstrengths of this paper

• This paper proposed a new model structure which significantly reduce the model computation time.

• The proposed methods have been applied well-established model structures in speech.

• The model shows good performance on some speech tasks.

Major weaknesses of this paper:

• This paper essentially states that self-attention could be entirely replaced by SummaryMixing which may be considered as a MLP network. This is a claim that overthrows the modeling development trend in the speech community, which is from MLP to RNN and then Transformer. Although the authors use experiments to support the claim, it is still not easy to understand why such simple architecture is enough for speech modeling. We must be very cautious to examine such a claim. Therefore, more experiments need to be done for larger scopes.

• Although the authors have tried to address the novelty concern in the rebuttal, the proposed SummaryMixing still seems to be an incremental change from HyperMixer by comparing these two methods in Figure 3 of Appendix. Of course, the change can be applied to Conformer and Branchformer to replace self-attention. But the fundamental change is still the one from HyperMixer.

• As pointed out by several reviewers, there is a concern about how SummaryMixing performs on long-form speech: the mean vector becomes a mixture of more frames, degrading the ability to discriminate individual frames/tokens. It is nice to see the authors provided an additional experiment to show the superiority of SummaryMixing over self-attention for long-form speech. However, it is surprising to see self-attention degraded so much when the utterance length is around 30s in Figure 7 of Appendix. This contradicts with the modeling design of Whisper, the popular speech foundation model, which uses 30s chunk and the performance drops when smaller-size chunk is used.

• The experiments always use less than 1000 hours of speech data. In the speech community, the discovery with small scale data doesn’t always generalize. Furthermoer, the results on AMI test set already showed SummaryMixing clearly degraded the self-attention from 18.2% WER to 19.4%.

**Justification For Why Not Higher Score:**

There are too many concerns on this paper such as the novelty and experiment verification. As mentioned in the above meta review, the paper's claims cannot be supported well by current experiments.

**Justification For Why Not Lower Score:**

N/A

---

### Decision · Program_Chairs · 2024-01-16

Reject